# CYCLIN-B1/2 and -D1 act in opposition to coordinate cortical progenitor self-renewal and lineage commitment

Daniel W. Hagey [1,2✉], Danijal Topcic[1,2], Nigel Kee[1], Florie Reynaud[1], Maria Bergsland[1], Thomas Perlmann[1] & Jonas Muhr [1✉]

The sequential generation of layer-specific cortical neurons requires radial glia cells (RGCs) to precisely balance self-renewal and lineage commitment. While specific cell-cycle phases have been associated with these decisions, the mechanisms linking the cell-cycle machinery to cell-fate commitment remain obscure. Using single-cell RNA-sequencing, we find that the strongest transcriptional signature defining multipotent RGCs is that of G2/M-phase, and particularly CYCLIN-B1/2, while lineage-committed progenitors are enriched in G1/S-phase genes, including CYCLIN-D1. These data also reveal cell-surface markers that allow us to isolate RGCs and lineage-committed progenitors, and functionally confirm the relationship between cell-cycle phase enrichment and cell fate competence. Finally, we use cortical electroporation to demonstrate that CYCLIN-B1/2 cooperate with CDK1 to maintain uncommitted RGCs by activating the NOTCH pathway, and that CYCLIN-D1 promotes differentiation. Thus, this work establishes that cell-cycle phase-specific regulators act in opposition to coordinate the self-renewal and lineage commitment of RGCs via core stem cell regulatory pathways.

[1] Department of Cell and Molecular Biology, Karolinska Institutet, Solnavägen 9, SE-171 65 Stockholm, Sweden. [2] These authors contributed equally: Daniel W. Hagey, Danijal Topcic. ✉email: daniel.hagey@ki.se; jonas.muhr@ki.se

Projection neurons in the mammalian cortex are organised into six distinguishable layers with distinct molecular and functional properties. During cortical development, these neuronal subtypes are generated in a strict temporal order from multipotent radial glia cells (RGCs), which give rise to deep-layer neurons (layers V and VI) followed by upper-layers neurons (layers II–IV), and finally cortical glia[1]. While RGCs need to balance maintenance of the progenitor pool with generation of distinct cortical cell types, the genetic programs coordinating self-renewal and lineage commitment remain poorly understood.

During the initial stages of cortical development, neuroepithelial cells divide symmetrically to expand the number of progenitors in the ventricular zone[2]. At the onset of cortical neurogenesis, at E10.5 in mouse, these cells convert into RGCs[3,4], which have the capacity to undergo asymmetric divisions. This mode of division results in one RGC, to maintain the progenitor pool, and one daughter cell that commits to neurogenesis, either directly or via contribution to a transient population of proliferating intermediate progenitor cells (IPCs)[5–7]. One of the mechanisms by which asymmetric divisions result in commitment to neuronal differentiation is via the unequal inheritance of cell-fate determinants between daughter cells[8]. For instance, RGC daughter cells that inherit high levels of active NOTCH signalling components tend to remain as RGCs, while those with lower levels of NOTCH pathway activity commit to neurogenesis[7,9–11]. Gain- and loss-of-function studies have also demonstrated the transcription factor SOX2 to have key functions in regulating stem cell maintenance[12,13]. Consistent with this, SOX2 is asymmetrically inherited during cortical progenitor division, such that it is expressed at higher levels in RGCs compared with IPCs[14,15].

Experiments aiming to understand the mechanisms that regulate self-renewal and lineage commitment have provided evidence for an intimate relationship between these cellular decisions and the cell-cycle machinery[16]. For example, by regulating the length of the G1 phase in the cortex, CYCLIN-D1 has been implicated in controlling the onset of neurogenesis by promoting the formation of IPCs[17,18], whereas the ability of CYCLIN-D1 to promote neurogenesis in the spinal cord can be dissociated from its cell-cycle function[19]. In contrast, regulatory components of S, G2 and M phases have been shown to maintain human embryonic stem cells in a pluripotent state, while their absence has been suggested to make the G1 phase permissive to lineage commitment[20–22]. Although these findings indicate an important role for cell-cycle components in cell-fate decisions, their function in coordinating self-renewal with cortical progenitor differentiation remain poorly understood.

A general issue when examining self-renewal and lineage commitment during corticogenesis is the heterogeneity of progenitor cell-cycle and differentiation states. By performing single-cell RNA sequencing, we are able to distinguish cortical cells at different stages of commitment to deep- and upper-layer neurogenesis. This distinction allows us to identify differentially expressed cell-surface proteins, which we then utilise to isolate and analyse specific populations of multipotent and lineage-committed progenitors. Interestingly, these molecular and functional characterisations reveal key factors involved in the regulation of G2/M and G1/S cell-cycle phases as potential determinants of cell-fate commitment. Thus, we perform functional analyses in vivo to demonstrate that B- and D-type CYCLINs control the timing of cortical neurogenesis in an opposing manner by promoting RGC maintenance and lineage commitment, respectively. Furthermore, we provide evidence that the NOTCH pathway is a key target of CYCLIN regulation and that CDK1-associated kinase activity is important for CYCLIN-B1's ability to counteract cortical progenitor differentiation. Together, this work unveils essential pathways and molecular mechanisms linking the regulators of cell-cycle progression and neural stem cell differentiation.

## Results

**Single-cell RNA-seq reveals a cortical maturation axis**. To examine how lineage commitment decisions are regulated during corticogenesis, we first characterised cortical cells using single-cell RNA-seq. To capture proliferating progenitors, as well as those committed to deep- (layers V and VI) and upper-layer (layers II–IV) neurogenesis (Fig. 1a), we randomly collected and sequenced single cortical cells (549 cells in total) from mice at five embryonic stages, from embryonic (E) day 9.5 to E18.5 (Fig. 1b; Supplementary Data 1)[1]. Following quality control cell filtering, we used a workflow of t-distributed neighbour embedding–nearest neighbour (tSNE-NN) mapping, Infomap graph-based clustering[23,24] and weighted gene co-expression network analysis (WGCNA)[25] to uncover cell types and gene expression patterns amongst the sequenced cells. Although the most variably expressed genes in our data set separated Prom1+ progenitors from Dcx+ neurons, they were unable to reveal any further cell type diversity (Supplementary Fig. 1a). Thus, to limit our analysis to cells within cortical pyramidal neuronal lineages, gene sets specifically expressed by immune cells, glia or interneurons[26] (Supplementary Data 2) were used to identify and remove these cell types from further consideration (Supplementary Fig. 1b–e).

Next, we wished to assess the lineage relationships between progenitors and neurons with respect to their commitment to cortical lineage decisions (Supplementary Data 3). While both Monocle DDRTree[27] and our tSNE-NN map method were readily able to separate the neurons into distinct groups when informed with previously defined cortical layer identity genes[26] (Supplementary Fig. 2a, b; Supplementary Data 2), this gene set could not separate progenitors into molecularly distinct clusters. Therefore, we next informed our graph-based clustering algorithm with lineage non-specific genes involved in both deep- and upper-layer neurogenesis. To derive this gene set, we used PROM1 sorting to separate progenitors and neurons when deep- (E11.5 progenitors and E12.5 neurons) and upper- (E15.5 progenitors and E16.5 neurons) layer lineages are formed[28]. Bulk RNA sequencing and differential gene expression (DESeq2)[29] analysis identified genes differentially expressed by progenitors and neurons of both deep- and upper-layer lineages (Supplementary Fig. 2c; Supplementary Data 2). Together, these general differentiation and neuronal layer-specific gene sets produced a tSNE-NN map with clearly separated groups of RGCs (Prom1+), intermediate progenitors (Eomes+) and neurons (Dcx+) (Fig. 1c, d). Moreover, the differentiation stage of each cell within the tSNE-NN map was consistent with an independent quantification of cortical maturation stage, which was itself highly correlated with that derived by Monocle pseudotime (R-squared value 0.97) (see "Methods"; Fig. 1e, f; Supplementary Fig. 2d–i).

**Identification of temporally distinct cortical trajectories**. The well-established sequential generation of layer-specific neurons during cortical development prompted us to analyse if we could identify temporally distinct differentiation trajectories within our data set. Interestingly, examination of the Infomap cell cluster (Fig. 2a) maturation stages revealed that the clusters located on the inner surface of the graph had equivalents on the outer surface, with overlapping maturation-stage values (Fig. 2b). Despite this similarity, the Infomap clusters along the inner surface of the graph (Fig. 2a) were of significantly younger embryonic age than their maturation-stage pairs on the outer surface (Fig. 2b). Importantly, this pattern was not observed in cortical cells

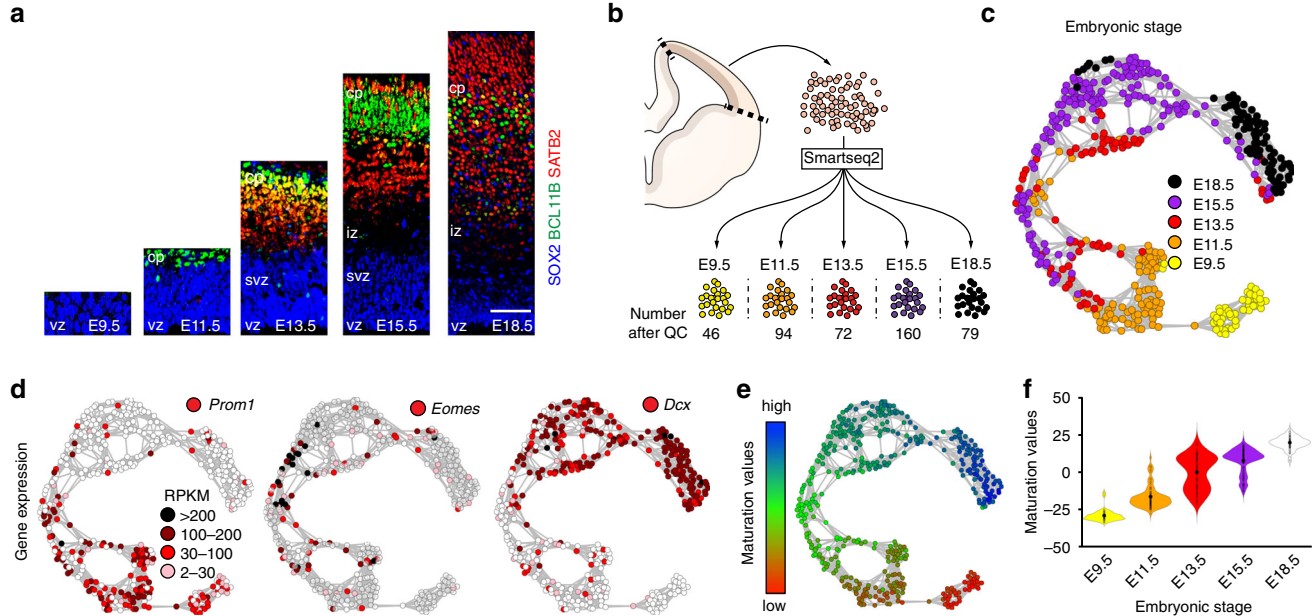

**Fig. 1 Single-cell RNA-seq reveals a cortical maturation axis. a** Immunohistochemistry shows expression of the progenitor markers SOX2, the deep-layer marker BCL11B and the upper-layer marker SATB2 in E9.5 to E18.5 cortices. Immunohistochemistry shown in (**a**) was repeated three times. **b** Experimental design for single-cell RNA-seq Smartseq2 analysis of cortical development. Number of cells sequenced; E9.5, 71; E11.5, 96; E13.5, 76; E15.5, 190; E18.5, 116. **c** tSNE-NN map of cortical cells passing quality control cell filtering. E9.5 (yellow), E11.5 (orange), E13.5 (red), E15.5 (purple) and E18.5 (black). **d** tSNE-NN maps show *Prom1, Eomes* and *Dcx* expression levels in the sequenced cells. **e** tSNE-NN map coloured by maturation values of sequenced cells, where red is low and blue is high. **f** Violin plots show the distribution of maturation-stage values for the cells sequenced from each embryonic age. These are inset by rings corresponding to the individual data points, a filled dot at the group mean and a vertical line showing standard error. Scale bar in (**a**) represents 24 μm E9.5, 30 μm, 36 μm E11.5, E15.5 and 54 μm E18.5. vz ventricular zone, svz subventricular zone, iz intermediate zone, cp cortical plate.

isolated at E9.5 (Infomap clusters red and magenta; Fig. 2a), or for three clusters of E11.5 cortical progenitors (Infomap clusters medium aquamarine, brown and yellow; Fig. 2a), which lacked maturation-stage pairs (Fig. 2b).

Since the age and maturation stage of the Infomap clusters were consistent with two temporally distinct cortical differentiation trajectories, we next examined if the different Infomap clusters corresponded with cells committed to deep- or upper-layer neurogenesis. Indeed, using single-cell differential expression (SCDE)[30] analysis to compare Infomap clusters of the early-differentiation trajectory (red in Fig. 2c) with those of the late-differentiation trajectory (blue in Fig. 2c), we found that known deep-layer genes (e.g., *Tbr1, Bcl11b, Foxp2, Tle4, Sox5* and *Fezf2*) were repeatedly enriched in the Infomap clusters of the early-differentiation trajectory, while canonical upper-layer genes (e.g., *Cux1, Cux2, Satb2, Unc5d, Pou3f2* and *Pou3f3*)[1,26,30,31] were more specific to clusters of the late-differentiation trajectory (Fig. 2d; Supplementary Fig. 2j; Supplementary Data 4). Moreover, by correlating each cell in the tSNE-NN map to deep- and upper-layer neuroblast gene profiles (Supplementary Data 4), we found that progenitors within each trajectory expressed the genes defining their own lineage's neurons more highly than those defining the opposite trajectory (Fig. 2e). Thus, the early- and late-differentiation trajectories identified appear to represent cortical cells undergoing deep- and upper-layer neurogenesis, respectively.

**Expression of lineage genes converges in uncommitted cells.** One plausible interpretation of the data is that the Infomap clusters lacking maturation-stage pairs, represent progenitors that are uncommitted to a specific cortical differentiation trajectory (green in Fig. 2c). To address this idea functionally, we next examined the neuronal cell-fate competence of cortical E9.5 neuroepithelial cells following forced differentiation in vitro for

48 h (Fig. 3a). Indeed, while approximately half of the resulting TUJ1[+] neurons expressed the deep-layer markers BCL11B (CTIP2) and SOX5, the other half expressed the upper-layer markers SATB2 and POU3F2 (BRN2) (Fig. 3b, c)[2]. Furthermore, E9.5 cell transcriptomes displayed a significantly greater correlation to the E11.5 cells lacking maturation-stage pairs than to E11.5 or E13.5 progenitors newly committed to deep- or upper-layer neurogenesis, respectively (Figs. 2b, 3d).

To uncover groups of gene that describe cortical cells as they commit to deep- and upper-layer trajectories, we next performed WGCNA on genes differentially expressed between Infomap clusters of similar maturation stages. This approach revealed groups of genes expressed during the earliest stages of commitment to cortical neurogenesis. To visualise the different gene sets, their average expression scores were plotted on the tSNE-NN map, and the in vivo mRNA expression patterns of representative genes were examined (Supplementary Fig. 3a, b)[31]. Interestingly, the gene sets enriched in cells at early-commitment stages to deep- or upper-layer neurogenesis (deep-layer-trajectory gene sets 1–2 and upper-layer-trajectory gene sets 1–2; Supplementary Fig. 3a, b) converged in uncommitted E11.5 cells, which co-expressed genes enriched in both differentiation trajectories (Fig. 3e, f; Supplementary Fig. 3a, b). Moreover, consistent with these cells being in an uncommitted state, they also co-expressed well defined deep- and upper-layer neuronal genes at levels that were resolved in cells newly committed to either differentiation trajectory (Supplementary Fig. 3c, d). Gene ontology (GO) analysis revealed that the genes shared by uncommitted E11.5 cells and cells newly committed to deep-layer neurogenesis (deep-layer-trajectory gene set 1) were enriched for terms associated with DNA replication and neuron maturation (Fig. 3g). In contrast, the genes shared by uncommitted E11.5 cells and E13.5 and E15.5 progenitors that contribute to upper-layer neurogenesis (upper-layer-trajectory gene set 1) were strongly enriched for terms involved in stem cell

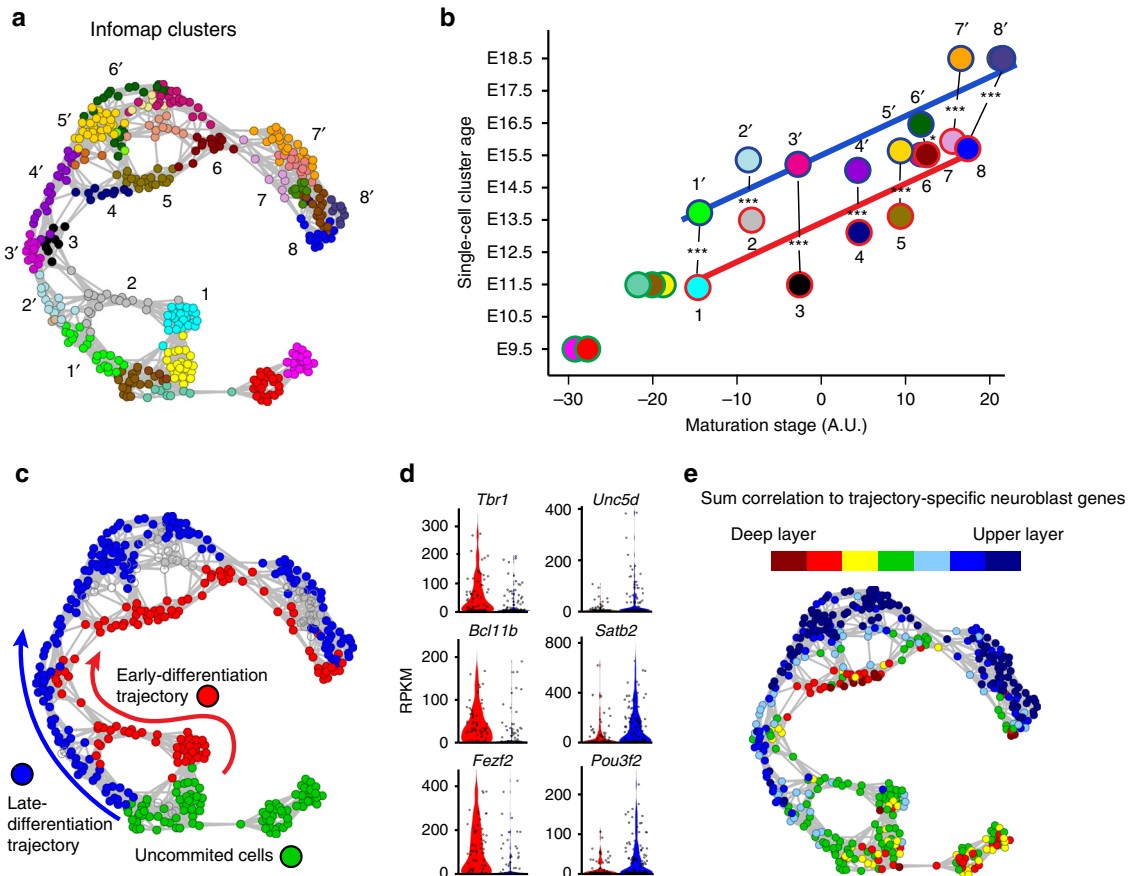

**Fig. 2 Identification of temporally distinct cortical differentiation trajectories. a** *t*SNE-NN map coloured by Infomap clusters. Numbering indicates maturation-stage matched clusters. **b** Numbered and colour-coded circles represent average values for Infomap clusters displayed in (**a**). The Infomap clusters are based on the following number of cells; cluster 1 25 cells, cluster 1´ 21 cells, cluster 2 19 cells, cluster 2´ 14, cluster 3 9 cells, cluster 3´ 14 cells, cluster 4 10 cells, cluster 4´ 21 cells, cluster 5 30 cells, cluster 5´ 21 cells, cluster 6 14 cells, cluster 6´ 20 cells, cluster 7 7 cells, cluster 7´ 21 cells, cluster 8 15 cells and cluster 8´ 8 cells. Infomap clusters are plotted based on their embryonic age and maturation-stage values, with best-fit lines for Infomap clusters of early-differentiation (red) and late-differentiation (blue) trajectories. *P*-values for the Infomap cluster comparisons; 1 vs. 1´ (1.34e-9), 2 vs. 2´ (1.38e-6), 3 vs. 3´ (6.63e-11), 4 vs. 4´ (1.35e-5), 5 vs. 5´ (6.30e-12), 6 vs. 6´ (1.59e-2), 7 vs. 7´ (9.65e-4), 8 vs. 8´ (2.32e-4). **c** *t*SNE-NN map depicting uncommitted cells (green), and cells committed to the early-differentiation (red) or late-differentiation (blue) trajectories. **d** Violin and dot plots of deep- (*Tbr1*, *Bcl11b* and *Fezf2*) and upper-layer gene expression (*Unc5d*, *Satb2* and *Pou3f2*) in cells of early- and late-differentiation trajectories, respectively. Analysis is based on 129 cells in early-differentiation trajectory, and 140 in late-differentiation trajectory. **e** *t*SNE-NN map showing sum correlation of each cell to deep- and upper-layer-trajectory neuron gene profiles. Red and blue colours indicate strong correlation to deep- and upper-layer-trajectory gene sets, respectively. Statistics based on two-tailed *t* tests; \**P* < 0.05, \*\*\**P* < 0.001.

maintenance and mitotic functions (Fig. 3g). Thus, the uncommitted cells we identify at E11.5 are unique in their similarity to multipotent E9.5 cells and their co-expression of deep- and upper-layer differentiation trajectory genes.

**Uncommitted cells have multipotent differentiation potential.** Based on their transcriptional profiles, we hypothesised that the uncommitted E11.5 cells we identified bioinformatically represent multipotent RGCs, with the potential to generate progeny of both deep- and upper-layer neuronal lineages. To functionally address this possibility, we next identified genes for cell-surface proteins that were predominantly expressed in uncommitted E11.5 cells (*Hmmr* and *Gpc6* enriched in upper-layer-trajectory gene set 1; *Ednrb* enriched in upper-layer-trajectory gene set 2) (Fig. 4a; Supplementary Fig. 4a–d), as well as cell-surface proteins predominantly expressed in E11.5 progenitors committed to deep-layer neurogenesis (*Slc1a5* enriched in deep-layer-trajectory gene set 1 and *Efna5* enriched in deep-layer-trajectory gene set 2) (Fig. 4a; Supplementary Fig. 4e, f). Via fluorescent-activated cell

sorting (FACS), antibodies against these cell-surface proteins were used to isolate cells from the total pool of *Sox2*-GFP+ cortical progenitors dissected from *Sox2*^EGFP^/+ mice[32] (Fig. 4b; Supplementary Fig. 4g–l). Notably, following 48 h of differentiation in vitro, we found that HMMR+, GPC6+ and EDNRB+ cells isolated from the E11.5, or E13.5 cortex, all showed a significantly greater propensity to generate upper-layer neurons (SATB2+, POU3F2+ and TUJ1+) at the expense of deep-layer neurons (BCL11B+, SOX5+ and TUJ1+), than the overall *Sox2*-GFP+ cell population depleted of these cell-surface proteins (Fig. 4c, d; Supplementary Fig. 5a–e). At E15.5, all *Sox2*-GFP+ progenitors generated upper-layer neurons, regardless of their expression of HMMR, GPC6 or EDNRB (Fig. 4c, d; Supplementary Fig. 5a–e). In contrast, cells isolated at E11.5 based on their expression of SLC1A5 or EFNA5 appeared lineage-committed and generated almost exclusively neurons expressing deep-layer markers, when differentiated in vitro for 48 h (Fig. 4e, f; Supplementary Fig. 5f–h).

SOX2 itself has been reported to be expressed at higher levels in cortical RGCs than in IPCs[14,15] and was represented in upper-layer-trajectory gene set 1 (Supplementary Fig. 3b). Consistent

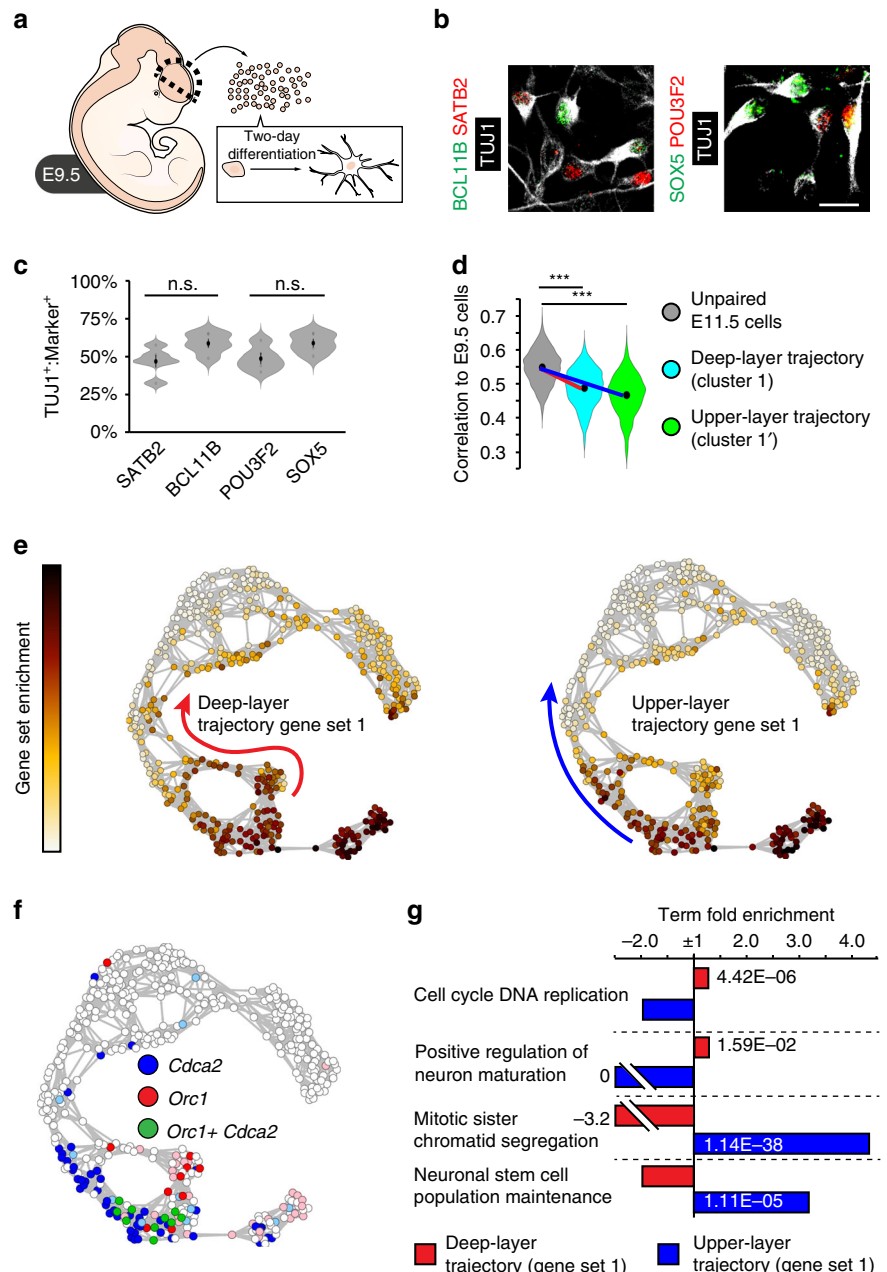

**Fig. 3 Lineage-specific gene expression converges in uncommitted progenitors. a** Experimental design for cortical in vitro differentiation assays.
**b**, **c** Immunohistochemistry (**b**) and quantifications (**c**) showing BCL11B, SATB2, SOX5 or POU3F2 expression in TUJ1+ neurons derived from E9.5 cortical cells after in vitro differentiation ($n = 5$ biological independent experiments). **d** Violin plots show correlations between the transcriptomes of E9.5 cells, uncommitted E11.5 cells (grey) or cells committed to deep-layer (cyan; red line; P-value < 1e-300) or upper-layer (green; blue line; P-value < 1e-300) differentiation trajectories. These correlations are based on 46 E9.5 cells 51 uncommitted E11.5 cells, 25 cells committed to deep-layer differentiation and 21 cells committed to upper-layer differentiation. **e** tSNE-NN maps coloured by WGCNA gene set enrichment scores for deep-layer-trajectory gene set 1 (red line) and upper-layer-trajectory gene set 1 (blue line). **f** tSNE-NN map coloured by Orc1 (from deep-layer-trajectory gene set 1; pink 20–100 RPKM, red >100 RPKM) and Cdca2 (from upper-layer-trajectory gene set 1; light-blue 60–100 RPKM, blue >100 RPKM) expression, with co-expressing cells coloured in green. **g** GO-term fold enrichments and P-values for genes in WGCNA deep-layer-trajectory gene set 1 (red) and upper-layer-trajectory gene set 1 (blue). Violin plots are inset by rings corresponding to the individual data points, a filled dot at the group mean and a vertical line showing standard error. Scale bar in (**b**) represents 10 μm. Statistics based on two-tailed t tests; ***P < 1e-300. P-values for the GO-term analysis are derived from a Binominal test. Source data are provided as a Source Data file.

with this, separation of E11.5 cortical cells based on their expression of *Sox2*-GFP revealed that cells expressing the highest levels of GFP, and thus SOX2 protein (Supplementary Fig. 6a, b), generated similar numbers of deep- and upper-layer neurons when cultured in vitro (Supplementary Fig. 6c, d). Furthermore, the competence of E11.5 progenitors to generate upper-layer

neurons decreased in parallel with SOX2 levels (Supplementary Fig. 6c, d). Together, these experiments functionally confirm our bioinformatic identification of multipotent RGCs and lineage-committed progenitors, which can be separated through their specific expression of cell-surface proteins, or by their distinct expression levels of the transcription factor SOX2 (Fig. 4g).

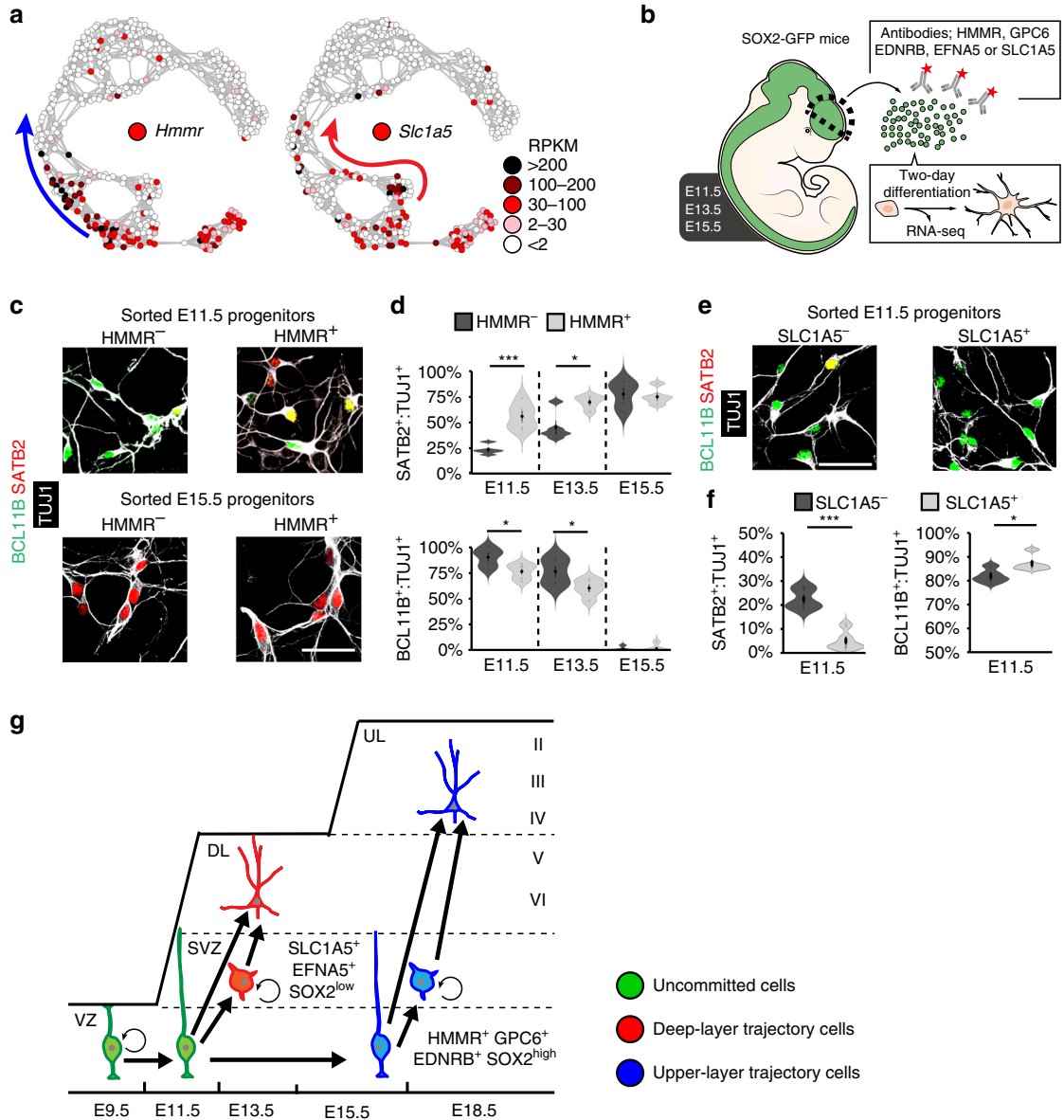

**Fig. 4 Uncommitted cortical progenitors display multipotent differentiation potential. a** tSNE-NN maps coloured by *Hmmr* or *Slc1a5* expression levels in cortical cells. **b** Schematic showing dissection of E11.5, E13.5 and E15.5 *Sox2*-GFP cortices, antibody labelling of cell-surface markers enriched in uncommitted progenitors and progenitors committed to upper-layer (HMMR and GPC6, enriched in upper-layer-trajectory gene set 1; EDNRB enriched in upper-layer-trajectory gene set 2), or deep-layer (SLC1A5, enriched in deep-layer-trajectory gene set 1 or EFNA5 enriched in deep-layer-trajectory gene set 2) differentiation trajectories, FACS-based isolation and in vitro differentiation or RNA-seq analysis. **c**, **d** Immunohistochemistry (**c**) and quantification (**d**) of BCL11B and SATB2 expression in TUJ1+ neurons derived from HMMR+ and HMMR− progenitors after two days of differentiation. E11.5 (*n* = 5 biological independent experiments; p-values SATB2 5.38e-4, BCL11B 0.023), E13.5 (*n* = 4 biological independent experiments; *P*-values SATB2 0.015, BCL11B 0.048) and E15.5 (*n* = 5 biological independent experiments). **e**, **f** Immunohistochemistry (**e**) and quantification (**f**) of BCL11B and SATB2 expression in TUJ1+ neurons derived from E11.5 SLC1A5+ and SLC1A5− progenitors after 2 days of in vitro differentiation (*n* = 5 biological independent experiments; *P*-values SATB2 1.48e-4, BCL11B 0.03). **g** Schematic of relationships between uncommitted cortical cells (green) and cortical cells committed to deep- (red) or upper-layer (blue) differentiation trajectories, labelled with markers analysed here. vz ventricular zone, svz subventricular zone, DL deep layer (layer V and VI), UL upper layer (layer II to IV). Scale bar in (**c**, **e**) represents 20 μm. Violin plots are inset by rings corresponding to the individual data points, a filled dot at the group mean and a vertical line showing standard error. Statistics based on two-tailed *t* tests; *\*P* < 0.05, \*\*\**P* < 0.001. Source data are provided as a Source Data file.

**Cell-cycle features separate RGCs from committed progenitors**. To further characterise the identified populations of RGCs and lineage-committed progenitors, we next performed bulk RNA sequencing on these cells directly after their isolation from E11.5 cortices (Fig. 4b). Hierarchical clustering and differential gene expression analysis revealed substantial molecular similarities between HMMR+, GPC6+ and EDNRB+ cells, which expressed

genes associated with RGCs, and separated these from lineage-committed EFNA5+ and SLC1A5+ cells committed to deep-layer neurogenesis (Fig. 5a, b; Supplementary Fig. 7a–d). Furthermore, gene set enrichment analysis (GSEA)[33] demonstrated that, in comparison with lineage-committed cells, genes overrepresented in RGCs were repeatedly enriched in pathways involved in the G2/M-phase transition, Notch signalling and the TGFβ signalling

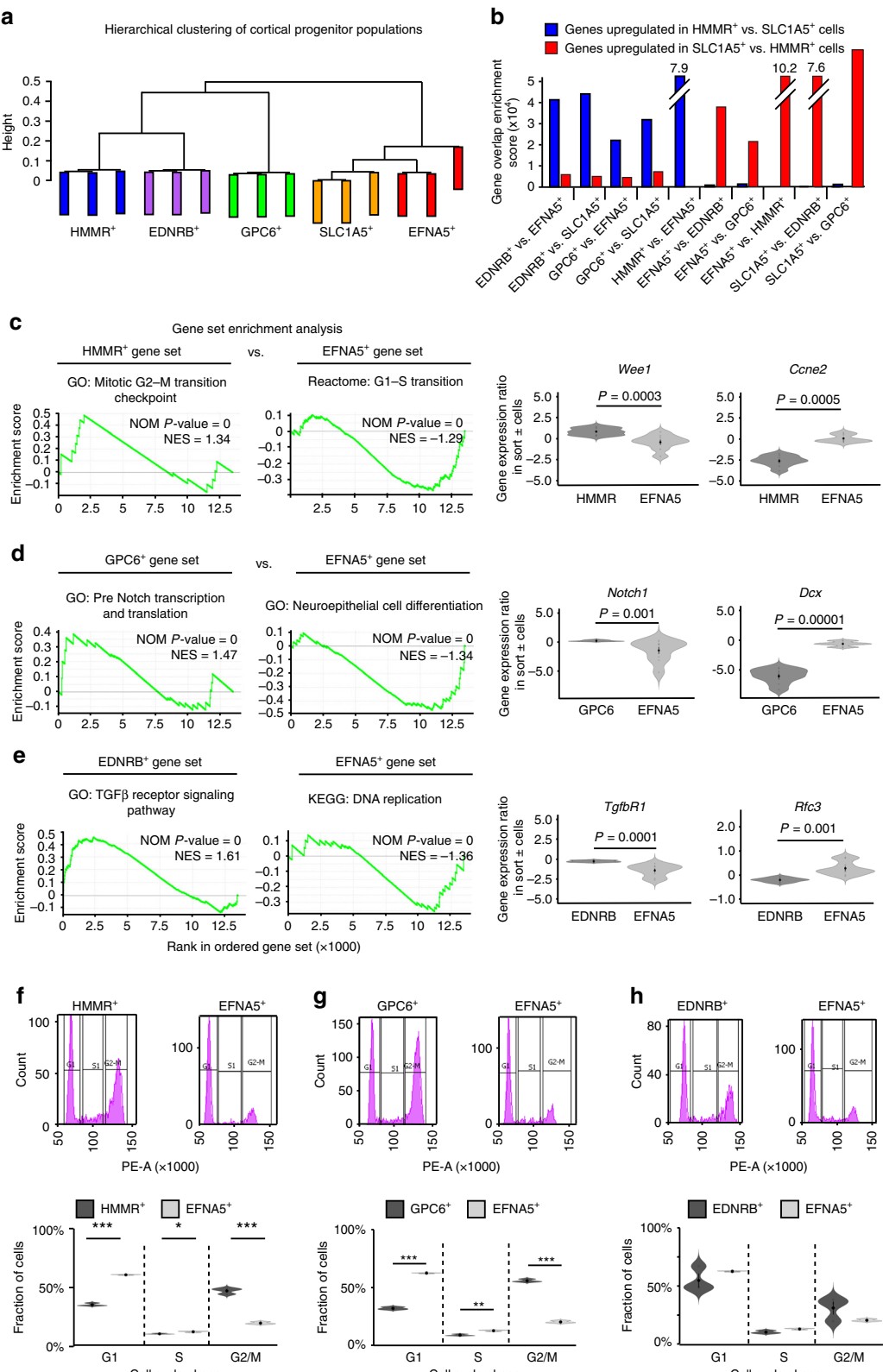

pathway (Fig. 5c–e; Supplementary Fig. 7e, f). In contrast, genes overrepresented in lineage-committed progenitors were associated with the G1/S-phase transition, DNA replication and cell differentiation processes (Fig. 5c–e; Supplementary Fig. 7e, f).

The connection between cortical progenitor competence and specific cell-cycle phases prompted us to determine the cell-cycle profiles of RGCs and lineage-committed progenitors. To proceed,

the DNA of these cell populations, sorted from E11.5 cortices, was labelled with propidium iodide (PI), so that their DNA content, and thus cell-cycle phase enrichment, could be determined using FACS. Consistent with the gene expression analysis above, HMMR+ and GPC6+ cells, and cells expressing high levels of Sox2-GFP (bin 4 cells), were significantly more likely to be found in G2/M phase than SLC1A5+ and EFNA5+

**Fig. 5 Cell-cycle features separate RGCs from committed progenitors. a** Hierarchical clustering on variable genes expressed >0.5 RPKM in sorted E11.5 HMMR[+], EDNRB[+], GPC6[+], SLC1A5[+] or EFNA5[+] cortical cell populations. **b** Bar graphs showing gene overlap enrichment of genes differentially expressed (according to DESeq2) between sorted E11.5 cortical HMMR[+] cells (blue bars) and SLC1A5[+] cells (red bars) and genes differentially expressed between the other sorted E11.5 uncommitted and committed cell populations. **c–e** GSEA terms enriched in HMMR[+] (**c**), GPC6[+] (**d**) and EDNRB[+] (**e**) E11.5 cortical cells when compared with EFNA5[+] progenitors. Violin plots show expression of genes relevant to the enriched terms displayed. *P*-values, nominal (NOM) *P*-values and normalised enrichment scores (NES) are indicated. **f–h** FACS analysis of propidium iodide-treated HMMR[+] cells (**f**) (*P*-values (vs. EFNA5[+] cells); G1 phase 3.27e−4, S phase 0.016, G2/M phase 7.67e−4), GPC6[+] cells (**g**) (*P*-values (vs. EFNA5[+] cells); G1 phase 1.8e−4, S phase 2.65e−3 and G2/M phase 5.15e−6), EDNRB[+] cells (**h**) and EFNA5[+] (**f–h**) E11.5 cortical cells. N = 8797 HMMR[+] cells, 11309 GPC6[+] cells, 3149 EDNRB[+] cells and 5341 EFNA5[+] cells over three independent experiments. Violin plots show the proportion of analysed cells in G1, S and G2/M cell-cycle phases (*n* = 3 biological independent experiments). Violin plots are inset by rings corresponding to the individual data points, a filled dot at the group mean and a vertical line showing standard error. Stars indicate significant differences between indicated groups based on two-tailed *t* tests; *P < 0.05, **P < 0.01 and ***P < 0.001. Source data are provided as a Source Data file.

cells, or cells expressing low levels of *Sox2*-GFP (bin 1 cells) (Fig. 5f–h; Supplementary Fig. 8a, b). These latter cell populations were instead more likely to be found in G1- or S phase (Fig. 5f–h; Supplementary Fig. 8a, b). Moreover, by determining the time BrdU-labelled E11.5 cells spent within M phase, as determined by PH3, we found that RGCs, as defined by high levels of SOX2 expression, entered M phase earlier and were more abundant overall within this cell-cycle phase than lineage-committed progenitors defined by lower expression levels of SOX2 (Supplementary Fig. 8c)[34].

**Cell-cycle phase-specific CYCLINs affect cell-fate decisions**. The finding that uncommitted and committed progenitors were enriched for cell-cycle phase-specific genes raised the question of whether these genes could be involved in regulating cortical neurogenesis. Interestingly, the cell-cycle regulators *Ccnb1* and *Ccnb2* (encoding for CYCLIN-B1 and -B2), which function during the M phase, were among the most significantly enriched genes in upper-layer-trajectory gene set 1 (Fig. 6a; Supplementary Data 5). Similarly, *Ccnd1* (encoding for CYCLIN-D1), which functions during G1, was among the most significantly enriched genes in deep-layer-trajectory gene set 1 (Fig. 6a; Supplementary Data 5). Thus, to address their potential roles in regulating deep- and upper-layer neurogenesis, we next modulated CYCLIN expression in E12.5 cortices using in utero electroporation (Fig. 6b). In comparison with GFP control electroporations, we found that overexpression of CYCLIN-B1 or -B2 (Supplementary Fig. 9a) led to a significant increase in the proportion of electroporated SATB2[+] and POU3F2[+] upper-layer neurons in the E18.5 cortex, at the expense of BCL11B[+] and SOX5[+] deep-layer neurons (Fig. 6c, d, Supplementary Fig. 9c–g). Consistent with this, decreasing the levels of *Ccnb1* and *Ccnb2*, through shRNA-mediated knockdown (*Ccnb1/2* shRNA-GFP; Supplementary Fig. 9b), increased the fraction of deep-layer neurons and decreased the number of upper-layer neurons when compared with the electroporation of an shRNA control (Fig. 6e, f; Supplementary Fig. 9f, g). In contrast to these results, overexpression of CYCLIN-D1, though not its homolog CYCLIN-D2 (Supplementary Fig. 9a, c, d), increased the generation of deep-layer neurons (Fig. 6c, d; Supplementary Fig. 9c-g), while its shRNA-mediated knockdown (*Ccnd1* shRNA-GFP; Supplementary Fig. 9b) decreased it (Fig. 6e, f; Supplementary Fig. 9f, g). Thus, while high levels of CYCLIN-D1 at E12.5 stimulated progenitors to generate deep-layer neurons, high levels of CYCLIN-B1/2 had the opposite effect and promoted progenitors to commit to upper-layer neurogenesis.

**CYCLIN-B1/2 and CYCLIN-D1 regulate lineage commitment**. Rather than having an instructive role in directing progenitors towards specific cortical fates, we hypothesised that CYCLIN-B1/2

and CYCLIN-D1 were influencing cell-fate decisions by modulating the time at which electroporated progenitors committed to neurogenesis. To address this possibility, we electroporated cortices at E14.5, a cortical stage when deep-layer neurogenesis is complete and upper-layer neurogenesis is ongoing[28]. At this stage of development, overexpression of CYCLIN-B1, or knockdown of *Ccnd1*, decreased the fraction of electroporated cells that became upper-layer neurons (Supplementary Fig. 10a–d). Instead, these manipulations led to an increased proportion of SLC1A3[+] astrocytes (Fig. 7a–c), a cell type that is generated following upper-layer neurogenesis[28]. Moreover, in these experiments, knockdown of *Ccnb1/2* or overexpression of CYCLIN-D1 promoted upper-layer neurogenesis at the expense of later-born astrocytes (Fig. 7a–c; Supplementary Fig. 10a–d). To further address the possibility that CYCLINs can affect the timing of neurogenesis, we next examined the formation of committed TBR2[+] IPCs 20 h after altering the levels of CYCLIN-B1/2 and CYCLIN-D1 in E12.5 cortices. Consistent with the results above, while overexpression of CYCLIN-B1, or knockdown of *Ccnd1*, reduced the fraction of electroporated cells expressing TBR2 (Fig. 7d, e), knockdown of *Ccnb1/2* expression, or overexpression of CYCLIN-D1, had the opposite effect and increased it (Fig. 7d, e). These results suggest that, rather than promoting specific cell fates, CYCLIN activity regulates corticogenesis by affecting the commitment to differentiation (Fig. 7c).

The spindle orientation of dividing cells is one important mechanism controlling RGC maintenance and differentiation[7]. To determine if CYCLINs could be involved in regulating this process, we sequenced mRNA from cortical progenitors 20 h after modulating their CYCLIN levels at E12.5 (Fig. 7f). By examining the cumulative effect of CYCLIN-B1 overexpression and *Ccnb1/2* knockdown on gene expression (see "Methods"), we found several key factors promoting symmetric divisions[35] to be positively regulated by CYCLIN-B1 (Fig. 7g). Consistent with these findings, overexpression of CYCLIN-B1 for 20 h increased the division angle of electroporated cells with reference to the ventricular surface, whereas knockdown of *Ccnb1/2* decreased it (Fig. 7h, i). Notably, these effects on spindle orientation were achieved without a consistent influence on the cell-cycle phase dynamics of the electroporated cells (Supplementary Fig. 10e, f).

**CYCLIN-B1/2 delay lineage commitment via NOTCH activation**. To examine how CYCLINs regulate the differentiation of cortical progenitors, we further analysed the sequencing data we obtained after altering CYCLIN expression in E12.5 cortical progenitors (Fig. 7f). Notably, a comparison with the gene expression profiles of sorted RGCs and progenitor cells committed to deep-layer neurogenesis revealed that, while high levels of CYCLIN-B1 upregulated genes enriched in E11.5 HMMR[+] cells, CYCLIN-D1 upregulated genes that were overrepresented in SLC1A5[+] cells (Supplementary Fig. 11a). When considering the

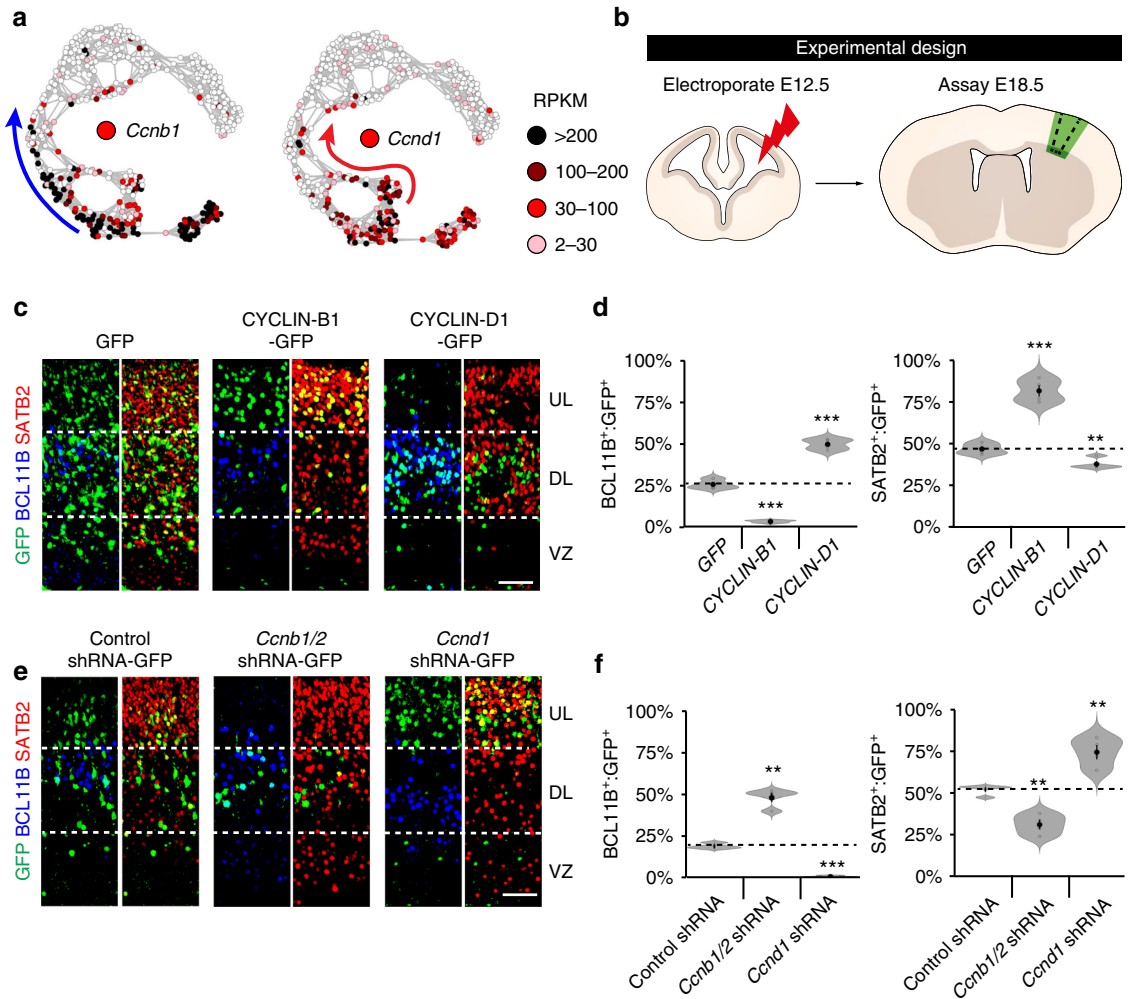

**Fig. 6 Cell-cycle phase-specific CYCLIN's affect cortical cell-fate decisions. a** *t*SNE-NN maps coloured by the expression levels of *Ccnb1* or *Ccnd1*.
**b** Schematic of cortical electroporations performed at E12.5 and processed for immunohistochemistry at E18.5. **c**–**f** Immunohistochemistry and
quantification of BCL11B (blue) or SATB2 (red) expression in cortical cells electroporated with vectors expressing GFP (green) either alone or together with
CYCLIN-B1 (*P*-values (vs. GFP); BCL11B 2.33e-4, SATB2 5.5e-4) or CYCLIN-D1 (*P*-values vs. GFP; BCL11B 3.25e-5, SATB2 8.48e-3) (**c**, **d**), control shRNA,
*Ccnb1/2* shRNA (*P*-values (vs. GFP); BCL11B 1.21e-3, SATB2 6.87e-3) or *Ccnd1* shRNA (*P*-values (vs. GFP); BCL11B 2.13e-4, SATB2 4.43e-3) (**e**, **f**) (*n* = 4
biological independent experiments). Scale bar in (**c**, **e**) represents 40 μm. Violin plots are inset by rings corresponding to the individual data points, a filled
dot at the group mean and a vertical line showing standard error. Stars indicate significant differences between indicated groups based on two-tailed *t* tests;
***P* < 0.01 and ****P* < 0.001. Source data are provided as a Source Data file.

cumulative effects of overexpression and knockdown on gene
expression, we found that CYCLIN-B1 strongly repressed *Ccnd1*
expression, whereas CYCLIN-D1 mildly upregulated *Ccna1*,
*Ccnb1* and *Ccnb2* (Supplementary Fig. 11b). On a genome-wide
scale, GSEA of these experiments demonstrated that high levels of
CYCLIN-B1, or knockdown of *Ccnd1*, upregulated genes impli-
cated in regulating the G2/M-phase cell-cycle transition, NOTCH
and TGFβ signalling (Fig. 8a, b; Supplementary Fig. 11c, d). This
was in comparison to CYCLIN-D1 overexpression, or knock-
down of *Ccnb1/2*, which increased the expression of genes
involved in the induction and progression of S phase, as well as
Myc target genes (Fig. 8a, b; Supplementary Fig. 11c, d).

NOTCH signalling has well-documented functions in stem cell
maintenance and delaying the onset of neurogenesis[36]. Since
CYCLIN-B1 regulated the expression of genes in the NOTCH
pathway (Fig. 8b), we wished to examine its function in
corticogenesis, downstream of CYCLIN-B1. We found that
misexpression of the NOTCH intracellular domain (NICD) in
E12.5 cortices mimicked CYCLIN-B1 overexpression and
increased the proportion of upper-layer neurons at the expense

of deep-layer neurons (Fig. 8c, d; Supplementary Fig. 12a, b). In
contrast, blocking NOTCH activity at this time, through the
electroporation of a dominant negative form of mastermind
(dnMM), forced deep-layer neurogenesis (Fig. 8c, d; Supplemen-
tary Fig. 12a, b). Importantly, NICD promoted upper-layer
neurogenesis even when *Ccnb1/2* was knocked down, and dnMM
forced deep-layer neurogenesis in the presence of co-
electroporated CYCLIN-B1 (Fig. 8c, d; Supplementary Fig. 12a,
b). Moreover, cortical progenitors electroporated with NICD
at E14.5 primarily generated SLC1A3[+] astrocytes instead of
SATB2[+] upper-layer neurons (Supplementary Fig. 12c–f),
whereas forced expression of dnMM at this stage induced
upper-layer neurogenesis (Supplementary Fig. 12c–f). Together,
these results indicate that the ability of CYCLIN-B1 to regulate
cortical neurogenesis is dependent on the downstream activation
of NOTCH signalling.

**CDK1 acts with CYCLIN-B1/2 to regulate cortical neurogen-
esis.** To gain further insight into the potential mechanisms by

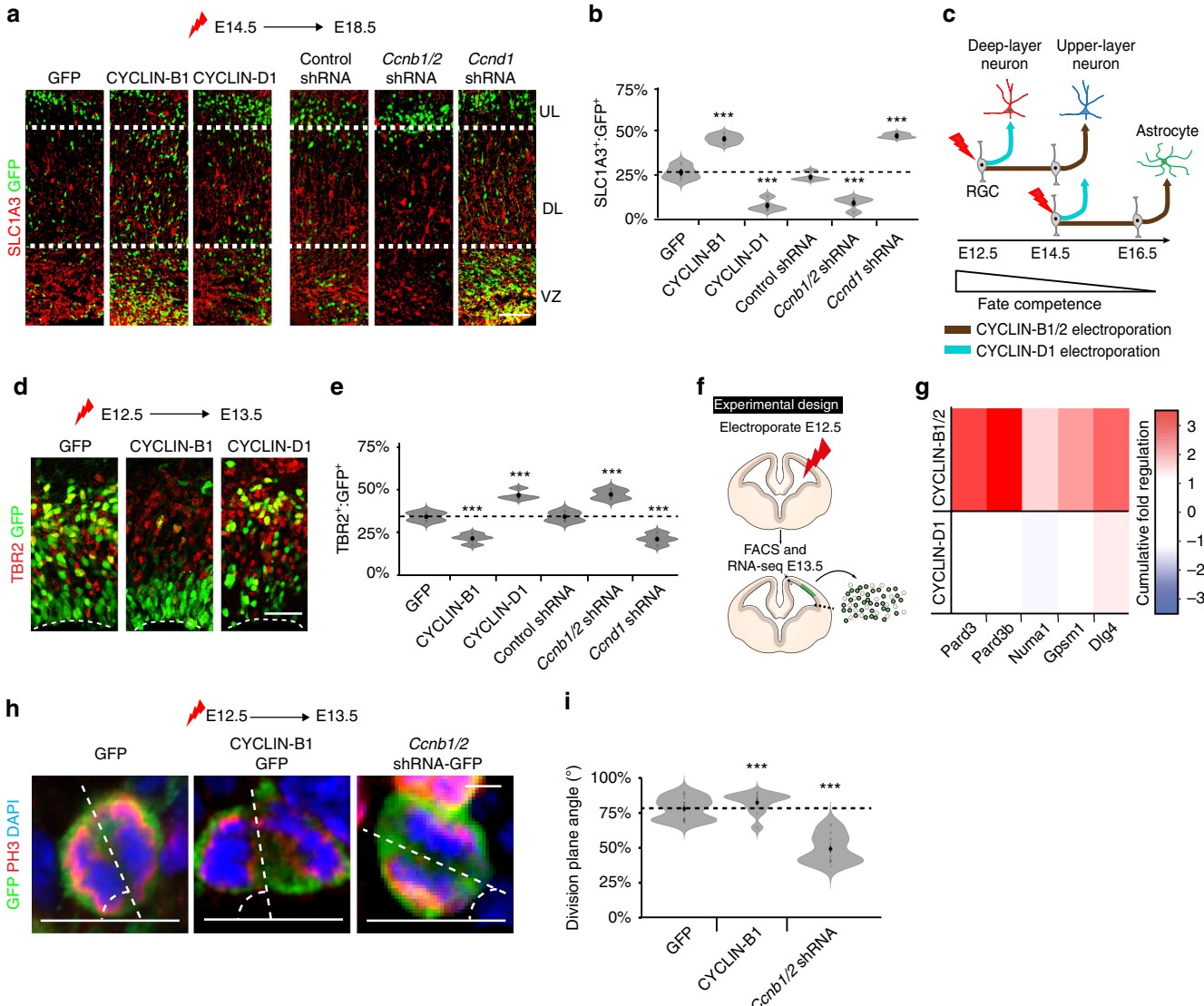

**Fig. 7 CYCLIN-B1/2 and CYCLIN-D1 regulate cortical progenitor differentiation. a, b** Analysis of cortical electroporation experiments performed at E14.5 and assayed at E18.5. Immunohistochemistry (**a**) and quantifications (**b**) show expression of the astrocyte marker SLC1A3 (red) in cells electroporated with vectors expressing GFP (green) either alone or together with CYCLIN-B1, CYCLIN-D1, control shRNA, *Ccnb1/2* shRNA or *Ccnd1* shRNA (n = 4 biological independent experiments). *P*-values SLC1A3 expression (vs. GFP); CYCLIN-B1 4.08e-4, CYCLIN-D1 4.34e-4, *Ccnb1/2* shRNA 6.66-4, *Ccnd1* shRNA 9.08e-4). **c** Schematic model summarising electroporation experiments conducted in E12.5 and E14.5 cortices, in which CYCLIN-B1/2 and CYCLIN-D1 levels were manipulated. Brown arrows represent trajectories followed by CYCLIN-B1/2 electroporated cells and blue arrows represent trajectories followed by CYCLIN-D1 electroporated cells. **d, e** Analysis of cortical electroporation experiments performed at E12.5 and assayed at E13.5. Immunohistochemistry (**d**) and quantifications (**e**) show expression of TBR2 (red) in cells electroporated with vectors expressing GFP (green) either alone or together with CYCLIN-B1, CYCLIN-D1, control shRNA, *Ccnb1/2* shRNA or *Ccnd1* shRNA. (n = 4 biological independent experiments). *P*-values TBR2 expression (vs. GFP); CYCLIN-B1 2.19e-4, CYCLIN-D1 8.14e-4, *Ccnb1/2* shRNA 6.1-4 and *Ccnd1* shRNA 8.61e-4. **f** Schematic of cortical cells electroporated with vectors expressing GFP (green) either alone or together with CYCLIN-B1, CYCLIN-D1, *Ccnb1/2* shRNA or *Ccnd1* shRNA at E12.5 were isolated by FACS, and processed for RNA-seq at E13.5. **g** Assessment of the cumulative effect of CYCLIN-B1 or CYCLIN-D1 overexpression and knockdown on genes promoting symmetric divisions. **h** Immunohistochemistry of E13.5 cortices showing GFP expression (green), PH3 (red) and DAPI (blue) in mitotic cells following electroporation with vectors expressing GFP (n = 10 biological independent experiments) either alone or together with CYCLIN-B1 (n = 15 biological independent experiments) or *Ccnb1/2* shRNA (n = 14 biological independent experiments) at E12.5. White stippled lines indicate the division plane angle measured for each mitotic cell. **i** Quantification of division plane analysis of electroporated cells described in (**h**). *P*-values division plane angle (vs. GFP); CYCLIN-B1 1.61e-4 and *Ccnb1/2* shRNA 2.87-7. Scale bars represent 50 μm in (**a**), 30 μm in (**d**) and 2 μm in (**h**). Violin plots are inset by rings corresponding to the individual data points, a filled dot at the group mean and a vertical line showing standard error. Stars indicate significant differences between indicated group and GFP based on two-tailed t tests, with ***P < 0.001. Source data are provided as a Source Data file.

which CYCLIN-B1/2 promote RGC maintenance, we manipulated the activity of its primary kinase partner—CDK1, which was highly enriched within upper-layer-trajectory gene set 1 (Supplementary Data 5). We found that, similar to the activity of CYCLIN-B1, overexpression of CDK1 at E12.5 promoted upper-layer

neurogenesis at the expense of deep-layer identities. Conversely, a kinase-dead form of CDK1 (dnCDK1)[37] had the opposite function and promoted deep-layer neurogenesis (Fig. 9a, b). Furthermore, while knockdown of *Ccnb1/2* completely blocked the ability of CDK1 to promote upper-layer fates and resulted primarily in the

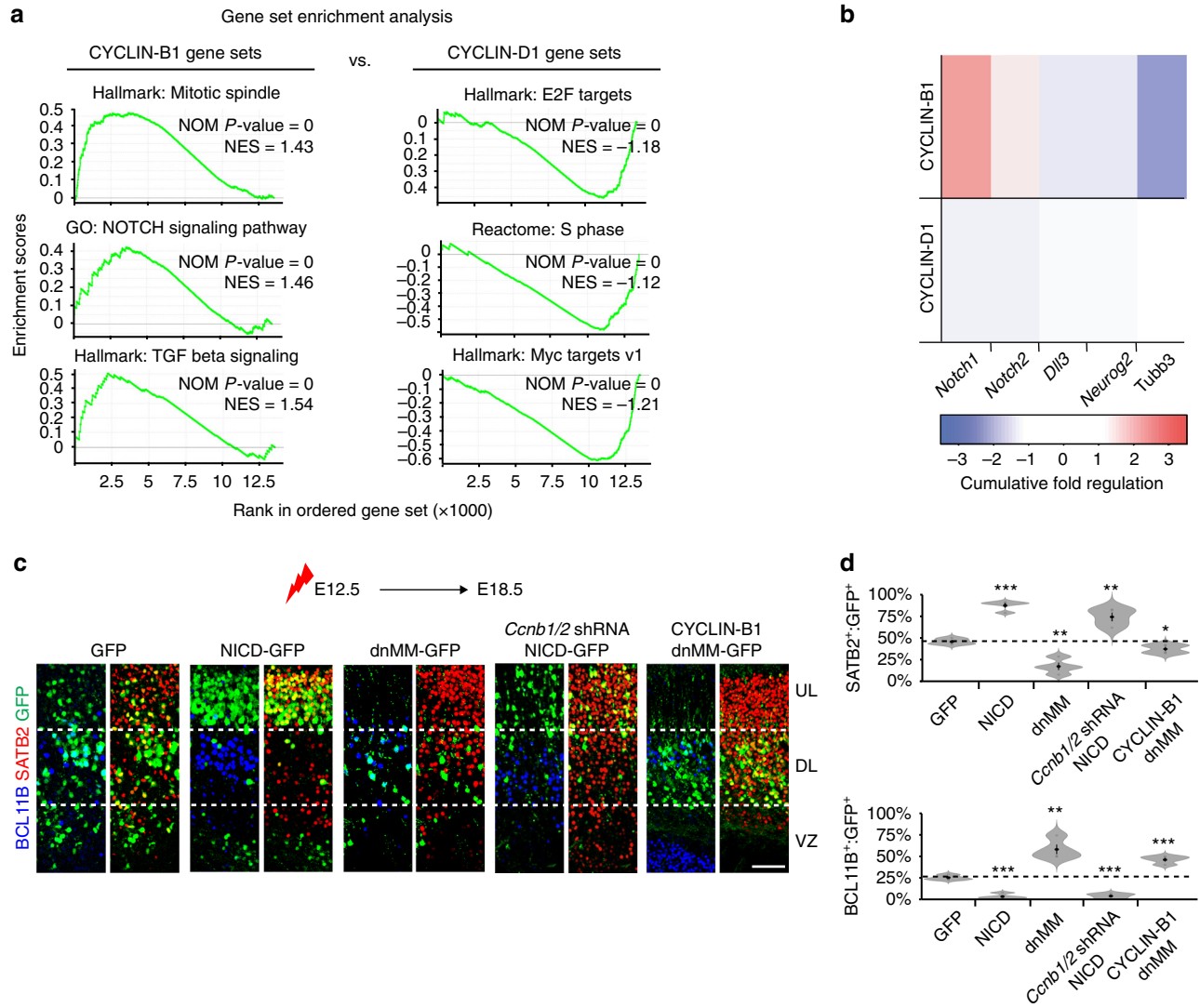

**Fig. 8 CYCLIN-B1/2 delay lineage commitment via NOTCH activation. a** GSEA terms enriched when comparing E13.5 cortical cells electroporated with CYCLIN-B1 versus CYCLIN-D1 for 24 h. **b** Cumulative effect on *Notch1*, *Notch2* and other genes implicated in neurogenesis, after overexpression and knockdown of CYCLIN-B1 or CYCLIN-D1 in cortical cells for 24 h. **c**, **d** Immunohistochemistry (**c**) and quantification (**d**) of GFP+ cells (green) expressing BCL11B (blue) or SATB2 (red) in E18.5 cortices after misexpressing GFP alone or together with NICD, dnMM, NICD/*Ccnb1/2* shRNA, or dnMM/CYCLIN-B1 (green) at E12.5 (*n* = 4 biological independent experiments). *P*-values BCL11B expression (vs. GFP); NICD 5.1e-5, dnMM 8.15e-3, *Ccnb1/2* shRNA/NICD 5.2e-5, CYCLIN-B1/dnMM 4.71e-4, *P*-values SATB2 expression (vs. GFP); NICD 7.4e-5, dnMM 3.91e-3, *Ccnb1/2* shRNA/NICD 7.51e-3, CYCLIN-B1/dnMM 0.035. Scale bar represents 40 μm in (**c**). Violin plots are inset by rings corresponding to the individual data points, a filled dot at the group mean and a vertical line showing standard error. Stars indicate significant differences between indicated group and GFP based on two-tailed *t* tests, with **P* < 0.05; ***P* < 0.01 and ****P* < 0.001. Source data are provided as a Source Data file.

generation of deep-layer neurons (Fig. 9a, b), dnCDK1 was only able to block the ability of CYCLIN-B1 to induce upper-layer neurogenesis by ~40% (Fig. 9a, b). Thus, although it is possible that CYCLIN-B1 also regulates cortical neurogenesis via kinase-independent mechanisms, our data support an important role for CDK1 activity in RGC maintenance (Fig. 9c).

## Discussion
The sequential generation of appropriate numbers of layer-specific cortical neurons requires RGCs to precisely balance self-renewal and differentiation, but how these cellular processes are coordinated has remained elusive. To gain insight into this question, we used single-cell RNA sequencing to identify and characterise the most prominent transcriptional signatures of

multipotent RGCs and progenitors committed to deep-layer neurogenesis. By using these data to guide our functional experiments, we have arrived at three major conclusions: Firstly, we find that gene expression profiles associated with G2/M and G1/S cell-cycle phases are amongst the most striking characteristics of RGCs and lineage-committed progenitors, respectively. Secondly, our data supports a model whereby specific cortical neuron subtypes arise from a single, progressively lineage-restricted, population of RGCs. Finally, we reveal that, while CYCLIN-D1 promotes the commitment of progenitors to cortical neurogenesis, CYCLIN-B1/2 and CDK1 maintain RGCs in an undifferentiated state by activating NOTCH signalling.

In order to confirm the relevance of our single-cell RNA-seq data with functional lineage analyses, we searched our layer-trajectory gene sets for cell-surface markers that could facilitate

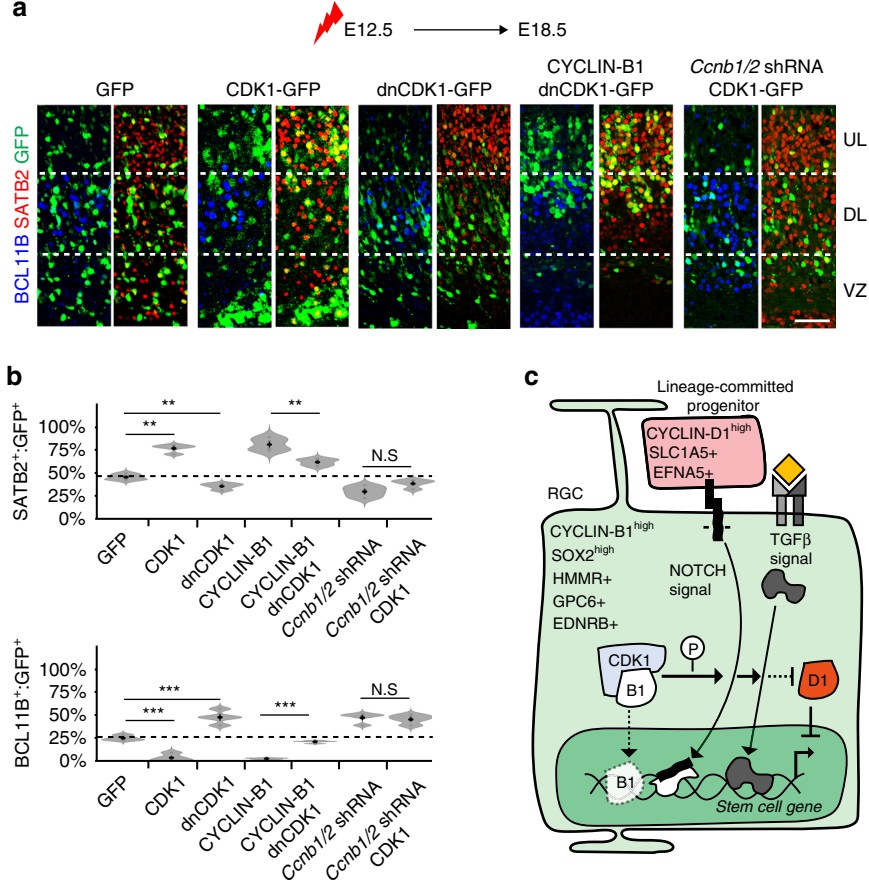

**Fig. 9 CDK1 acts together with CYCLIN-B1/2 in regulating corticogenesis. a, b** Immunohistochemistry (**a**) and quantification (**b**) of GFP⁺ cells (green) expressing BCL11B (blue) or SATB2 (red) in E18.5 cortices after misexpressing GFP alone or together with CDK1, dnCDK1, CYCLIN-B1, CYCLIN-B1 and dnCDK1, *Ccnb1/2* shRNA or *Ccnb1/2* shRNA and CDK1 at E12.5 ($n = 4$ biological independent experiments). *P*-values BCL11B expression (vs. GFP); CDK1 3.17e-4, dnCDK1 4.8e-3 and CYCLIN-B1/dnCDK1 6.78e-6. *P*-values SATB2 expression (vs. GFP); CDK1 5.7e-5, dnCDK1 5.17e-3 and CYCLIN-B1/dnCDK1 5.6e-3. Violin plots are inset by rings corresponding to the individual data points, a filled dot at the group mean and a vertical line showing standard error. Stars indicate significant differences between indicated groups based on two-tailed *t* tests, with **$P < 0.01$ and ***$P < 0.001$. **c** Model summarising features of RGCs and lineage-committed progenitors identified in this work. According to the model, the kinase activity of CDK1/CYCLIN-B1, and possibly gene regulatory functions on the chromatin level, activate stem cell-regulatory pathways, including those of NOTCH and TGFß, which delay the commitment of RGCs to cortical lineage differentiation. Scale bar represents 40 μm in (**a**). Source data are provided as a Source Data file.

FACS-based separation of multipotent RGCs from more committed progenitors in the E11.5 cortex. Through this approach, we identified endothelin receptor B (EDNRB), glypican proteoglycan 6 (GPC6) and hyaluronan-mediated motility receptor (HMMR) as markers of RGCs, whereas EFNA5 (EPHRIN A5) and SLC1A5 (solute-carrier family 1 member 5) were identified as markers for more committed cortical progenitors. Moreover, consistent with previous reports[14] SOX2 expression was also more highly expressed in RGCs in our data set and could be sorted for using *Sox2*^EGFP/+ mice. Importantly, several of these genes are not only markers of RGCs but have been previously described to play roles in neural stem cell biology. For instance, SOX2 has been shown to act in a concentration-dependent manner to regulate features of RGCs and committed progenitors in the developing cortex[15]. Neural precursors from mice lacking HMMR function display impaired symmetric divisions as well as a significant reduction in RGC numbers[38]. Similarly, EDNRB has been shown to be necessary for neural progenitor proliferation[39]. Moreover, in a previous Drop-seq based study, examining transcriptional differences between progenitors at different stages of cortical development, EDNRB was identified as one of a limited set of genes whose expression defined RGCs during corticogenesis and that continued to be expressed in stem cells in the

ventricular–subventricular zone of the adult forebrain[40]. While, the expression of *Efna5* and *Slc1a5* demarks committed progenitors, it is interesting to note that cortical cells expressing these genes are generally devoid of *Eomes* expression (compare for instance Fig. 1 and Supplementary Fig. 4). Thus, our findings support the notion that cortical progenitor commit to neurogenesis before they upregulate the expression of the IPC marker *Eomes*, for instance by reducing NOTCH signalling and upregulating *Neurog2*.

Importantly, the results of the aforementioned study agree with ours, and revealed that a major difference between the cortical progenitor clusters identified at E11.5 was their expression of cell-cycle phase-specific genes, as well as their enrichment in G2/M or S-phases[40]. The striking enrichment of multipotent RGCs in G2/M phase is a key finding and may be the result of several factors. For instance, symmetrically dividing RGCs may have G2/M-phase-enriched cell-cycle dynamics. In support of this, the M phase of committed Tis21⁻ RGCs in the cortex has been shown to be two-thirds longer than that of progenitors committed to neurogenesis (Tis21⁺ cells)[34]. Our data agree with this finding, as we demonstrate that cells expressing high levels of SOX2 spend significantly more time in the G2/M phase, compared with cells expressing lower levels of SOX2 and that both HMMR⁺ and

GPC6$^+$ cells are biased to G2/M phase. Alternatively, it is possible that these proteins are specifically upregulated during G2/M phase. Although we cannot exclude this possibility, it has been demonstrated that SOX2 levels do not vary significantly during the cell cycle[41]. However, in contrast to the above findings it should be noted that pharmacological-induced prolongation of the M phase has been shown to promote RGCs to commit to neurogenesis[42]. Nevertheless, the proteins we identify are not only markers of RGCs but also yielded key insights into mechanisms that may be of significance in neural stem cell maintenance.

Heterochronic transplantation[43,44], lineage tracing[45,46] and in vitro culture experiments[47] indicate that the precise correlation between neuronal birthday and subtype identity relies on the cell-fate competence progression of RGCs[10]. However, cortical neuronal diversity has also been suggested to depend on co-existing subpopulations of RGCs, with distinct fate potentials[48,49]. Here, we characterise a population of E11.5 RGCs that cannot be preferentially linked to deep- or upper-layer neurogenesis by our bioinformatic analysis. Consistent with this, these progenitors generated similar numbers of deep- and upper-layer neurons in an in vitro differentiation assay when we isolated them based on their expression of RGC cell-surface markers or *Sox2*-GFP expression levels. In contrast, when RGCs were isolated at E13.5 or E15.5, these cells were progressively more restricted to upper-layer neurogenesis. Thus, these results are compatible with a model by which a population of multipotent RGCs becomes progressively fate restricted with time. Notably, although forced expression of CYCLIN-B1 prevented cortical RGC differentiation, it did not preserve their multipotent cell-fate competence. Instead, the electroporated cells underwent normal lineage progression and committed to later formed cortical lineages in an appropriately timed manner. Moreover, we and others have obtained similar results by inhibiting cortical neurogenesis through NICD over-expression[50]. Although it is not fully understood how fate progression of RGCs is regulated, cell-cycle progression has been demonstrated to be dispensable for these cell-fate transitions[51]. However, the loss of Eed, the regulatory subunit of PRC2 that regulates chromatin accessibility, has previously been shown to accelerate fate competence progression of RGCs[52]. Moreover, heterochronic transplantation of RGCs into younger cortices causes these progenitors to re-enter a competence state typical of RGCs contributing to deep-layer neurogenesis[53]. One possible explanation for this temporal plasticity is the high levels of Wnt signalling components in younger cortices[53].

It is well understood that cell-cycle progression regulates many cellular processes beyond those required for cell division. For instance, CYCLIN-D1 has been previously described to induce an IPC state in the cortex[17,18] and promote neurogenesis in the spinal cord[19], which is consistent with our bioinformatic and functional experiments. In contrast, experiments on human embryonic stem cell (hESC) differentiation have shown that perturbing progression of S- and G2-phases promotes pluripotency maintenance. These authors found that the link between stem cell maintenance and specific cell-cycle phases was due to the stabilisation of p53 and activation of the TGFß pathway during S-phase perturbation, and upregulation of CYCLIN-B1 when G2 was delayed[20]. Analogous to our findings, they demonstrated that knockdown of *CCNB1* led to the suppression of pluripotency markers, while overexpression of CYCLIN-B1 maintained hESCs in a pluripotent state[20]. Hence, the functional link between CYCLIN-B1 and the maintenance of stem cell features is conserved between mouse RGCs and hESCs, which suggests that this may be a widely exploited relationship in diverse stem cell populations.

Apart from modulating the kinase activities of CDKs, CYCLIN-D1 has also been found to play a role in the nucleus, whereby it regulates the transcription of genes such as *NOTCH1*, by attracting and regulating the activity of coactivators and corepressors[22,53]. Whether CYCLIN-B1 acts via a similar mechanism to regulate genes implicated in stem cell maintenance is not currently established. However, we found that CDK1 is partly dispensable for CYCLIN-B1s ability to maintain cells in a RGC state. Thus, though CYCLIN-B1 is likely to act via CDK1 in regulating RGC differentiation, one interpretation of these results is that it may also act in a non-canonical, kinase-independent manner (Fig. 9c). We further demonstrate that one important mechanism by which CYCLIN-B1 regulates the timing of corticogenesis is by upregulating the expression of genes involved in NOTCH signalling. While we can only speculate how CYCLIN-B1 mediates this function, it is notable that high levels of CYCLIN-B1 reduce the expression of CYCLIN-D1, which has been shown to modulate the expression of NOTCH by interacting with its cis-regulatory regions directly[54].

In this paper, we have used an unbiased approach to uncover the roles, mechanisms and target pathways of CYCLINs in balancing progenitor self-renewal and lineage commitment during corticogenesis. Although we cannot comment on the relative influence of cell-cycle dynamics in RGC maintenance, our data strongly indicate that the expression levels of different CYCLINs are of key importance. Moreover, the wide spread employment of CYCLINs, and the molecular pathways they regulate, make it reasonable to hypothesise that CYCLINs also play an essential role in the regulation of stem cell differentiation in niches outside the developing nervous system.

## Methods

**RNA sequencing.** CD1 mouse cortices (9 cortices from E9.5, 12 cortices from E11.5, 10 cortices from E13.5, 9 cortices from E15.5 and 6 cortices from E18.5) were dissected and dissociated using a Miltenyi Neural Tissue Dissociation Kit (#130-092-628). Cells were directly picked under a dissection microscope using a mouth pipette and microcapillary into lysis buffer (0.2% Triton-X, 1U/µl RNAse inhibitor). For bulk RNA sequencing, RNA was extracted from sorted cells lysed in RLT buffer using a QIAGEN RNA purification kit (#74034). cDNA for bulk and single cells was prepared using the Smartseq2 protocol[55], while sequencing libraries were prepared using the Nextera XT kit according to the manufacturer's instructions. Sequencing reactions were performed on an Illumina HiSeq 2000.

**Sequence alignment, gene expression quantification and quality control.** Fastq files were aligned to the mm9 genome using Star v2.5. Gene expression was then assessed for each cell using rpkmforgenes. To filter unreliable data, cells with <200,000 mapped reads or 1500 < detected genes > 9000, were removed from the data set. Following this, hierarchical clustering using the R package hclust was performed on all cell-cell Spearman correlations based on the most variable genes within the data set, and any cell with an average correlation coefficient >3 standard deviations above or below the mean was removed from the data set. This removed 45 cells from further analysis.

**tSNE-NN-mapping method.** Different gene lists were used as input into the R (3.5.3) package Rtsne 0.15 (Supplementary Data 3) to create weighted PCA scores for each gene across the first 70 principle components, which were then projected into five dimensions. Within this space, the Euclidean distance between each cell's ten nearest neighbours was used to construct an adjacency matrix with edges weighted according to [(graph max Euclidean distance − pairwise Euclidian distance)/graph max Euclidian distance], while the reciprocal edge weight was also recorded. The directed weighted adjacency matrix was then visualised using the R package igraph 1.4.2.1 to create a force directed graph network of all cells in the data set. In such a graph, cells that have closely related transcriptomes will cluster together and create a network of connections that displays the continuum of relatedness between all cells in the data set.

**Gene sets for tSNE-NN mapping and non-cortical cell filtering.** First, we utilised the most variable genes expressed above rpkm 0.5 across the entire data set to look at the largest differences in our data set. This clustered cells on their differentiation stage but could not identify rare cell types in our data set. Thus, we utilised the cells previously identified as astrocytes ("Astro1" and "Astro2"), oligodendrocytes ("Oligo1", "Oligo2", "Oligo3", "Oligo4", "Oligo5" and "Oligo6"), interneurons

("Int1", "Int2", "Int3", "Int4", "Int5", "Int6", "Int7", "Int8", "Int9", "Int10", "Int11", "Int12", "Int13", "Int14", "Int15" and "Int16"), immune ("Mgl1", "Mgl2", "Pvm1" and "Pvm2"), ependymal ("Epend"), mural ("Peric" and "Vsmc") or endothelial cells ("Vend1" and "Vend2"), and isolated the top 50 genes describing each cell type by performing SCDE 2.14.0 between the specific groups and all other cells in their data set. These genes were used to inform the tSNE-NN map. Using WGCNA 1.67 on these same genes, we could highlight Infomap clusters that were enriched for glial, interneuron or immune gene expression patterns. After removing the two immune, 18 glial and 33 interneuron cells from our data set, we then used the same strategy to derive cortical layer genes from the same single-cell data set's upper-layer pyramidal neurons ("S1PyrL23") and deep-layer pyramidal neurons ("Sub-Pyr", "S1PyrDL", "S1PyrL4", "S1PyrL5", "S1PyrL5a", "S1PyrL6" and "S1PyrL6b"). Using a tSNE-NN map, these genes were readily able to segregate our single cells based on differentiation stage and the upper- or deep-layer identity of the differentiated cells, but was unable to resolve the diversity within our progenitor populations.

In order to describe the differences in gene expression between early and late progenitors, we performed bulk RNA-seq on PROM1-positive populations from the E11.5 and E15.5 cortex, as well as Prom1-negative populations from the E12.5 and E16.5 cortex. Using DESeq2 1.26, we derived genes with $P$-adjust < 0.01 and fold change >4 in both E11 PROM1$^+$ vs E12 PROM1$^-$ and E15 PROM1$^+$ vs E16 PROM1$^-$ experiments. Combining this list of genes with the previously described list of genes describing layer identity[26] allowed us to arrive at our final graph, showing the specific relationships between different stem cell populations and their fate specified progeny.

**Maturation-stage assignment and Monocle comparison.** PCA was performed using prcomp on all the pyramidal cortical cells in our data set based on a list of classical differentiation genes[56]. PC1 and PC2 were then rotated such that the number of detected genes correlated directly with PC1, allowing each cell to be given an individual maturation score and rank for its position along the linear differentiation scale of PC2. This scale matched very well with the expression of other genes known to describe cellular differentiation and maturation. Pseudotime (DDRTree) was performed using Monocle 2.14.0, with cell pseudotime ranks correlated to those from our maturation-stage assignments.

**Clustering, SCDE and WGCNA in situ.** In order to derive lineage genes from our data set, we clustered our cells based on our previously derived graph using Infomap community detection within the R package igraph. We then performed SCDE on pairs of clusters within two maturation-stage units of one another, resulting in comparisons between clusters within the early- and late-differentiation trajectories of similar maturation stage. Any genes that were found to have a Stouffers $Z$ score (sourced from Wikipedia [https://en.wikipedia.org/wiki/Fisher's_method]) above 0.4 in more than one pairwise comparison for more than one cluster of neuroblasts (maturation stage >10; from dark blue in early-differentiation trajectory and late-differentiation trajectory) was assigned as an upper or deep-layer gene (Supplementary Data 4), with several displayed as violin plots in Fig. 2d.

For weighted correlation network analysis (WGCNA), all 6620 genes with a combined $Z$ score >10 across all pairwise SCDEs were used as input for the WGCNA R package. We then assigned genes to the gene set to which they showed the lowest Gaussian mixture model (GMM) $P$-value. To confirm that these genes showed similar expression patterns in vivo, we used the Allen Brain Atlas' Developing Mouse Brain database and found this to be true across all areas of the single-cell graph.

**Gene ontology analysis.** Gene ontology $P$-value scores were obtained for the complete GO biological function catalog from pantherdb.org. Fold enrichment scores were normalised to the fold enrichment of a given term for all the genes in all gene sets being analysed. For Supplementary Fig. 1e, these genes were derived from SCDE between the different clusters highlighted by WGCNA, while for Fig. 3g we used the genes from WGCNA deep- and upper-layer-trajectory gene set 1.

**Gene set enrichment analysis (GSEA).** Sorted cell population and electroporation experiments were compared as indicated using GSEA software from http://software.broadinstitute.org/. Hallmarks, GO biological processes and complete curated gene sets were analysed for enrichment. Enrichment curves, normalised enrichment scores and NOM $P$-values are presented.

**Immunohistochemistry and western blot.** Tissue was fixed overnight at 4 °C, cryoprotected in 30% sucrose overnight at 4 °C, embedded in OCT and cryosectioned to 12 μm. Stainings were performed according to Hagey and Muhr[15] using antibodies against SOX2 (Goat sc-17320, Santa Cruz, 1/200), Phospho-Histone H3 (mouse clone 3H10, Millipore, 1/1000), EOMES (Rabbit ab23345, Abcam, 1/1000), BCL11B (rat ab18465, Abcam, 1/1000), SATB2 (Rabbit ab92446, Abcam, 1/1000), SOX5 (Rabbit, Ludwig Institute for Cancer Research, Muhr laboratory, 1/500), POU3F2 (Goat sc-6029, Santa Cruz, 1/250), TUJ1 (Chicken ab41489, Abcam, 1/1000), GPC6 (Alexa Fluor-conjugated bs-2177R−A647, Bioss, 1/100), HMMR

(Alexa Fluor-conjugated bs-4736R-A647, Bioss, 1/100), EDNRB (Alexa Fluor-conjugated bs-2363R-A647, Bioss, 1/100), EFNA5 (Alexa Fluor-conjugated bs-6048R-A647, Bioss, 1/100) and SLC1A5 (Alexa Fluor-conjugated bs-0473-A647, Bioss, 1/100). Western blots were performed according to Kurtsdotter et al.[57] using antibodies against CYCLIN-B1 (mouse sc-245, Santa Cruz, 1/1000), CYCLIN-B2 (mouse sc-28303, Santa Cruz, 1/1000) and CYCLIN-D1 (Rabbit ab134175, Abcam, 1/20,000).

**Animals and in utero electroporation.** All animal procedures and experiments were performed in accordance with Swedish animal welfare laws authorised by the Stockholm Animal Ethics Committee: Dnr N249/14. Animals were housed at a temperature of 22 °C, 50% humidity and a 12/12-h light/dark cycle.

Sox2-GFP animals were obtained from the Jackson laboratory (B6;129S-Sox2$^{tm2Hoch}$/J), and embryos were isolated at E11.5, E13.5 or E15.5 from 8–12-week-old C57/Bl females. In utero electroporation was performed on CD1 mice at E12.5 or E14.5 by sedating the pregnant female on a heated pad using isofluorane and protecting her eyes using eye gel. After isolating embryos and injecting the cortex with pCIG or pcDNA6.2 vectors in 1 × PBS + Fastgreen at 50 mV were directed to the appropriate hemisphere, they were sutured back into the mother, who was given two doses of Buprenorfin anaesthetic over a 24 h period. Brains were then isolated at E13.5 for bulk RNA-seq or at E13.5 and E18.5 for fixation and immunohistochemical analysis.

**FACS sorting and differentiation of neural progenitor populations.** Sorting of cells isolated from Sox2-GFP embryo cortices were immunostained with specific cell-surface protein antibodies and subjected to FACS on a BD FACSAria III to obtain a double-positive population of >98% purity. FACS sorting for PI was performed on a FACSvantage/DiVa using BD FACSDiVa software. The antibody sorted populations ranged from 0.5% to 5% of the parent population, and at least 30,000 cells were obtained. These cells were then plated on coated glass slides with rubberised 10-mm wells overlayed. E9.5 cells had to be grown in proliferation media containing EGF and FGF for 6 h before changing to differentiation media. The cells were grown in neural differentiation media for 48 h before being fixed in 2% PFA for 30 min at room temperature. These were then stained with neuronal subtype-specific antibodies in blocking solution containing 0.2% Triton-X overnight at 4 °C.

For cell-cycle analysis, cells were fixed in 70% ethanol, treated with 100 μg/ml propidium iodide for 30 min at room temperate and processed for flow cytometry analysis.

**Upper- and deep-layer gene expression.** Normalised upper- and deep-layer gene expression in different progenitor populations (Supplementary Fig. 3c, d) was calculated by dividing the expressions of Fezf2, Sox5, Tbr1, Bcl11b and Zic3 (for deep-layer gene expression), or Satb2, Pou3f2, Pou3f3, Unc5d and Zbtb20 (for upper-layer gene expression) in each single cell by the average expression for each gene, and then averaging and plotting these normalised expressions as a violin plot for each single cell in the different clusters.

**Trajectory-specific neuroblast genes and sum correlation to meta-neurons.** SCDE was performed between early- and late trajectory neuroblast clusters, which were defined as any cluster with an average maturation-stage value >0. All genes with a cZ < −0.4 or >0.4 in at least two pairwise comparisons with a single cluster were considered overexpressed in that cluster. Any gene found to be overexpressed in two neuroblast clusters of the same trajectory was then considered to be a trajectory-specific neuroblast gene (Supplementary Data 4).

These genes were then used to create early- and late trajectory meta-neurons by giving all genes in the early- and late trajectory neuroblast gene lists a value of 1, respectively, and all other genes a value of 0. Correlations were then calculated between these meta-neurons and all the cells in our data set. Correlations to the late trajectory meta-neuron were multiplied by -1 and these were added to each cell's correlation to the early trajectory meta-neuron to give a sum value. Cells in Fig. 2e were then coloured by their sum correlations, such that dark blue < −0.02 < blue > 0 < sky blue > 0.04 < green > 0.07 < yellow > 0.08 < red > 0.1 < dark red.

**Gene set overlap enrichment.** Enrichment scores were calculated as (number of genes overlapping between two sets/(number of genes in gene set 1 × number of genes in gene set 2)).

**Cumulative cyclin gene regulation.** Average gene expression RPKMs for overexpressions and knockdowns were first compared to GFP or shRNA control RPKMs. Knockdown-effect magnitude was inverted before being multiplied with that of overexpression. The cumulative regulation of overexpression and knockdown of CYCLIN-B1 and CYCLIN-D1 are displayed as heatmaps for specific genes in Figs. 7g and 8b, and Supplementary Fig. 11b.

**M-phase length tracking.** In order to judge the M-phase length of cells expressing high and low levels of SOX2, we first injected 100 μg BrdU per gram of body weight into nine mice. At 1-h intervals, up to 9 h, the mice were sacrificed and embryos

fixed in 4% PFA overnight. After performing BrdU/PH3/SOX2 triple stains, SOX2 levels were assayed using ImageJ (version 1.52) and overlap with BrdU and PH3 were judged according to Hagey and Muhr[15]. Area under curve significance was tested by compiling all combinations of SOX2 high- and low-expressing PH3/BrdU data points, and judging their difference by a two-tailed *t* tests. The Hausdorff distance was calculated between average curves.

**Constructs**. Cyclin, NICD, dnMM, CDK1 and dnCDK1 expression constructs were cloned by PCR amplification from cDNA and ligation into pCIG expression vectors. shRNA constructs were designed and cloned into pcDNA6.2-GW/EmGFP-miR according to the manufacturer's instructions. Efficiency and specificity of shRNA constructs were examined in mouse P19 cells. Hairpin sequences were Ccnb1: ATAATGGACACAGTCATGTACGTTTTGGCCACTGACTGACGTACATGAGTGTCCATTAT, Ccnb2: TATTCTTCAAATCACTGGACAGTTTTGGCCACTGACTGACTGTCCAGTTTTGAAGAATA and Ccnd1: TGGAAATGAACTTCACATCTGGTTTTGGCCACTGACTGACCAGATGTGGTTCATTTCCA.

Efficiency and specificity of shRNA constructs were examined in mouse P19 cells as demonstrated in Supplementary Fig. 9.

**Reporting summary**. Further information on research design is available in the Nature Research Reporting Summary linked to this article.

## Data availability

All data used in this work are available under NCBI accession SRP132833. All data supporting the findings and custom code within this paper are available from the corresponding authors upon reasonable request. The source data underlying Figs. 3–9 and Supplementary Figs. 5, 6, 8–10 and 12 are provided as a Source Data file.

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

## Acknowledgements

We are grateful to Science for Life Laboratory, the National Genomics Infrastructure, NGI and Uppmax for providing assistance in massive parallel sequencing and computational infrastructure. We are grateful to Mattias Karlén for illustrations. This research was supported by grants from the Swedish Research Council, The Swedish Cancer Foundation and The Knut and Alice Wallenberg Foundation. Open access funding provided by Karolinska Institute.

## Author contributions

D.W.H and J.M. designed the experiments. D.W.H. and D.T. generated single cell and bulk cDNAs. D.W.H., M.B. and N.K. analysed the sequencing data. D.T. performed FACS-based isolation of cortical cells and analysis of their cell-cycle state. D.W.H. and D.T. performed the in vitro differentiation assays. D.W.H., F.R. and J.M. generated plasmid cDNAs. D.W.H. and F.R. performed in vivo electroporation studies. T.P. provided intellectual support. D.W.H. and J.M. wrote the paper.

## Competing interests

The authors declare no competing interests.
