## [Peer Review File · Nature Communications]

Reviewers' comments:

Reviewer #1 (Remarks to the Author):

Review Hagey et al.

The paper by Hagey et al performs single cell RNA-seq to characterize the cell populations using cluster analysis. Bulk RNA-seq was used to determine how cell cycle phases might be involved in the regulation of cell fate in the developing brain. The transcriptional signatures in RGCs and IPCs appear to represent genes of distinct cell cycle phases, where RGCs express high levels of G2/M phase cell cycle proteins, while IPCs express G1 cyclins like D1. The authors then manipulate the expression of cyclins using in utero electroporation and find that manipulation of the expression of cell cycle proteins impact the fate of embryonic neural precursor cells.

Overall, this is a very nice study that addresses a very interesting question. The results are very novel, and the ss-RNA-seq is a great approach to characterize the transcriptome in the developing brain. I have some further suggestions to strengthen the paper:

1. The ssRNA-seq figures showing the trajectories of differentiation on the tSNE plots are very nice. The total number of cells used for ssRNA-seq seems rather low, 549. This is very surprising considering the embryo has a relatively large number of NPCs. A larger number of cells would strengthen the analyses, as some of the clusters contain relatively very few cells.
2. The section 'Uncommitted cortical progenitors display multipotent differentiation potential in vitro', is less exciting and does not add much to the story, thus I would recommend shortening this section up.
3. The in utero electroporation is an ideal approach by which to manipulate the embryonic brains, and the results obtained are clear and convincing, except the section examining division angle. To properly assess mode of division, ie symmetric vs asymmetric, a marker should be included that is asymmetrically segregated to more definitively demonstrate changes in the mode of division. Otherwise, this section should be removed, as it is not conclusive.
4. The authors should elaborate a bit more as to the mechanism by which CyclinB1 regulates notch signaling. What is the mechanism by which notch signaling is regulated by Cyclin B1.

Reviewer #2 (Remarks to the Author):

Hagey et al. investigated the transcription signatures of multipotent and lineage-committed progenitor cells. They identified multiple markers of the two neural progenitor populations including key factors involving cell cycle regulation. They further showed that B- and D-type cyclins control the timing of neurogenesis by promoting neural stem cell maintenance or differentiation, respectively. A detailed investigation identified that NOTCH signaling mediates the effects of B-type cyclins.

General comments:

Overall, I find the paper potentially important and interesting. Cell cycle has long been thought to be critical for the balance between neural stem cell maintenance and differentiation. This manuscript shows the novel involvement of B-type cyclin in regulating cortical neural progenitor cells. More

importantly, the authors showed that the effects of cyclins are independent of cell-cycle phase dynamics, which is intriguing and unexpected. Cellular and histological investigations are relatively thorough and comprehensive. However, the single-cell analysis has not been done properly.

Specific comments:

1. For the tSNE figures in Figure 1 and Supplementary Figure 1, the way the authors perform dimension reduction, data visualization, and cell filtering is bizarre. Generally, for a high-quality single-cell RNA seq analysis, radial glia, IPCs, interneurons, excitatory neurons, and microglia should be readily identified and separated by an unbiased dimensional reduction using the most variable genes. The fact that the authors fail to separate these cell types using this way suggests either low-quality data or a too small sample size. The authors end up using a defined set of genes to perform dimension reduction which makes the figures biased towards the hypothesis.

2. The lineage analysis in Figure 2 is also strange. First of all, a tSNE plot is not most ideal to visualize lineages. Have the authors tried UMAP? UMAP is supposed to perform better in terms of consistent and meaningful organization of cells and to better visualize lineages. Also, instead of visually identify the differentiation trajectories on the tSNE plot, the authors should do this in a more sophisticated way using Monocle DDRTree, Slingshot, or RNA velocity.

3. Also in Figure 2, when the authors define early and late differentiation trajectory, they based that on the expression of deep-layer and upper-layer genes. There are two problems with this. First, as mentioned in my second point, the authors should not assume a lineage relationship simply based on the proximity of clusters on a tSNE plot. For instance, it could be that cluster 2' gives rise to both clusters 3 and 3'. Second, the expression of the layer markers should be analyzed in each cluster separately, not in the whole "trajectory". Otherwise, the higher expression of these markers in more mature neurons will dominate the results and lead to wrong conclusions for the progenitor cells.

4. The authors describe the use of a directed approach in order to examine the lineage relationship between cells. To do so, they use Prom1+ as a sole marker to FACS-sort committed and uncommitted progenitor populations from different cortical developmental timepoints for downstream bulk RNA-seq. Why was Prom1 chosen for this analysis? Why were different timepoints used to sample Prom1 negative and positive populations (E11.5 and 15.5 for Prom1+ and E12.5 and 16.5 for Prom1-)? Dissecting the intricate trajectory relationships of progenitor cells based on bulk-RNA sequencing of cells sorted by a single marker (at two timepoints) seems highly biased. The authors should explain the rationale for this approach.

5. The authors hypothesize that the uncommitted E11.5 cells represent multipotent RGCs with the potential to generate both deep and upper-layer neuronal lineages. To test this, they chose cell-surface proteins enriched in upper-layer gene sets for FACS-sorting. It seems that these genes are also highly expressed in E11.5 newly committed upper-layer neurogenesis clusters (Fig. 2a, b; Fig. 3d). How can the authors be sure that FACS sorting is not selecting E.11.5 cells that are already committed rather than uncommitted E.11.5 cells as stated? These experiments show the propensity of E.11.5 cells to generate upper layer neuron identity at the expense of deep layer identity in vitro, but have not shown the reciprocal experiment (generation of deep layer identity at the expense of deep-layer). Thus, there is a possibility that the in vitro conditions favor the generation of upper-layer identity regardless of cell potential.

6. In Figure 4, are SOX2 levels of HMMR+, GPC6+, and EDNRB+ cells higher than SLCA5+ and EFNA5+ cells? If SLCA5+ and EFNA5+ cells are IPCs committed to deep-layer neurons, have the authors checked if they are TBR2+?

7. Also in Figure 4, the authors should be clear about the terms they are using to describe cell populations. In the present manuscript, they are very vague and confusing. For example, on page 11, lines 2-3, the authors write "...both multipotent and committed progenitors (respectively referred to as RGCs and IPCs from this point on)...". Are committed progenitor cells all IPCs? How about older radial glia cells that only contribute to upper-layer neurogenesis? The authors should be very clear about the lineage (deep-layer) and the developmental time point (E11.5) of the committed progenitors they are referring to. They should also extend this clarification to the following paragraphs.

8. From Figure 4 back to Figure 1, if the E11.5 deep layer committed progenitor cells are indeed IPCs, why are they not EOMES/TBR2 positive in Figure 1? Does it infer the low quality of the single-cell RNA seq data?

Minor points:

9. In figure 6c, 6e and Supp. Fig7 c,e,f. for some images it seems that SATB2 or POU3F2 are expressed from the VZ all the way through to the upper cortical layer. At E18.5 they should be restricted to the CP region. Is this an antibody specificity issue or are regions of cortex mislabeled?

10. There are several citation oversights. The function of cyclin D in promoting neurogenesis has been reported before by the Anderson lab 1. The ability of NOTCH intracellular domain to keep progenitors as proliferating radial glia cells has also been reported previously 2. These findings should be properly cited.

References:

1. Lukaszewicz, A. I. & Anderson, D. J. Cyclin D1 promotes neurogenesis in the developing spinal cord in a cell cycle-independent manner. *Proc. Natl. Acad. Sci. U. S. A.* 108, 11632–11637 (2011).
2. Mizutani, K. I. & Saito, T. Progenitors resume generating neurons after temporary inhibition of neurogenesis by Notch activation in the mammalian cerebral cortex. *Development* 132, 1295–1304 (2005).

Reviewer #3 (Remarks to the Author):

In this study, Hagey and collaborators use single-cell RNA sequencing to investigate the mechanisms linking the cell-cycle machinery to cell-fate commitment.

They find that RGCs are enriched in G2/M-phase genes, such as CYCLIN-B1, while IPCs highly express G1/S-phase genes, such as CYCLIN-D1. They also find cell-surface markers capable of isolating RGCs and IPCs. Finally, they demonstrate that CYCLIN-B1/2 cooperate with CDK1 to maintain RGCs in an uncommitted state by activating the NOTCH pathway, in contrast to CYCLIN-D1 which promotes differentiation. They conclude that cell-cycle phase-specific regulatory components act in opposition to coordinate the self renewal and lineage commitment of RGCs by controlling core stem cell regulatory pathways.

While these core findings are interesting (the findings of Fig. 4 are particularly nice) and represent a considerable amount of work, the manuscript is laid out in an unnecessarily convoluted manner and the figures and analyses are very poorly laid out / presented / performed. The authors should go back to their manuscript and carefully weigh which elements they think are most important to convey their message. Figures should be re-designed in depth.

Concerns:

1. Page 6 Line 6: "To achieve this, we performed bulk RNA sequencing on PROM1+ progenitors from E11.5 and E15.5 cortices and differentiating PROM1- cells from E12.5 and E16.5 cortices." Why did the authors do these experiments at different ages? They could have taken PROM1+ and PROM1- cells from the same cortices at E11.5 and E15.5 for instance.

2. Page 6 Line 12: "As visualized in the tSNE-derived nearest neighbour map, this analysis effectively separated cells according to their age and expression of the progenitor marker Prom1, the intermediate progenitor marker Eomes and the neuronal marker Dcx (Fig. 1c, d)." Why was bulk RNA-seq used to identify these markers? There are already known markers.

3. The method used to define the maturation stage value is unclear. The authors should provide a clearer and more comprehensive explanation. "Maturation stage assignment". The authors should provide references or list the genes used.
"PC1 and PC2 were then rotated such that the number of detected genes correlated directly with PC1": What does it mean? What is PC1 (=time)? This should be carefully justified.

4. "Furthermore, E9.5 cells displayed significantly greater gene expression correlations to the E11.5 cells from the Infomap clusters lacking maturation stage pairs than to E11.5 progenitors newly committed deep- or upper-layer neurogenesis (Fig. 2a, b; Fig. 3d)." What is the relevance of this statement?

5. Regarding Fig. 3, description of the bioinformatic analysis is very convoluted. Why was WGCNA used, what are the gene set 1 and 2? The authors should refer to a Sup Table here. In f, where do Cdca2 and Orc1 emerge from?

6. Slc1a5+, Efn5+, Sox2low cells are IPs. Given the tSNE in Fig. Sup. 4e, f, how many of the collected cells co-express Slc1a5 and Efn5 in the scRNAseq? Given the tSNE in Fig.1 representing Eomes expression (a classical IP marker) it seems that most of these cells do not express Slc1a5 and Efn5; how do the authors explain this? What is the number of cells that co-express these markers?

7. Regarding Fig. 5 and the gene set enrichment analysis, it is hard to understand how the authors found the GO. Are they at the top of the list? It might be nice to have a ranked list of GO. In 5c, the displays are just impossible to understand for non-specialists. The authors should re-think this and other panels.

8. Page 11 Line 20: "To proceed, RGCs and IPCs sorted from E11.5 cortices were labelled with propidium iodide (PI) and their cell cycle phase enrichments were determined using FACS." The authors should explain in the text how PI works (that the intensity of the fluorescence detected by the FACS will determine the cell cycle phase). Related to Fig 8b. Why show Neurog2 and Tubb3? There is no reference of this result in the text (see Page 15).

9. Related to Fig.9c. IPCs communicate with RGCs through NOTCH signaling. Which data show this? Other concerns (in order of appearance in the text)

1. Page 3 Line 7: "and finally cortical glia, 1.": please remove ",".

2. In the introduction, a coma is missing between references in the text.

3. Page 14 Line 12: "(Fig. 7d,)". e is missing.

4. In Fig. 8c and Sup Fig. 10, it would be nice to show the images for GFP control.

5. Discussion Page 19: "Heterochronic transplantation...", "Thus, these results are consistent with a model by which a population of multipotent RGCs becomes progressively fate restricted with time." The authors should present and discuss the recent findings of Oberst et al, 2019 in this context. Also Telley et al, 2019, Pilaz et al, 2016, Pilaz et al 2009 are important contributions which are missing.

Referee #1

- 1) *The ssRNA-seq figures showing the trajectories of differentiation on the tSNE plots are very nice. The total number of cells used for ssRNA-seq seems rather low, 549. This is very surprising considering the embryo has a relatively large number of NPCs. A larger number of cells would strengthen the analyses, as some of the clusters contain relatively very few cells.*

We agree with the referee that a larger number of cells could have helped the analysis. However, the number of cells, which were sequenced relatively deep using the Smart-seq2 protocol, was sufficient to identify and characterize cortical progenitor populations of different maturation levels. We feel that adding more cells at this stage would be complicate the analysis, while adding little to the conclusions that we confirm functionally. However, in the revised manuscript we have included discussion around previous single-cell RNA-seq studies, which analysed greater numbers of progenitors derived from the developing cortex. Their results are in line with ours and strengthen our findings. In the revised manuscript these studies have been discussed on p. 19;

Moreover, in a previous Drop-seq based study, examining transcriptional differences between progenitors at different stages of cortical development, EDNRB was identified as one of a limited set of genes whose expression defined RGCs during corticogenesis and that continued to be expressed in stem cells in the ventricular-subventricular zone of the adult forebrain⁴⁰.

Importantly, the results of the afore mentioned study agree with ours, and revealed that a major difference between the cortical progenitor clusters identified at E11.5 was their expression of cell-cycle phase specific genes, as well as their enrichment in G2/M or S-phases⁴⁰."

- 2) *The section 'Uncommitted cortical progenitors display multipotent differentiation potential in vitro', is less exciting and does not add much to the story, thus I would recommend shortening this section up.*

With all due respect to the reviewer's opinion, we feel that the identification of differentially expressed cell surface markers, which enable the isolation of uncommitted and committed cortical progenitors, is a significant finding of this paper. Moreover, functionally confirming the cells developmental potential is an important confirmation of the descriptive bioinformatic analysis. However, to meet the referees request, we have reduced and reorganized Fig. 4 and the text describing it (see track changes p. 10-11), as well as included better arguments as to why we do these experiments in the revised manuscript.

- 3) *The in utero electroporation is an ideal approach by which to manipulate the embryonic brains, and the results obtained are clear and convincing, except the section examining division angle. To properly assess mode of division, i.e. symmetric vs asymmetric, a*

marker should be included that is asymmetrically segregated to more definitively demonstrate changes in the mode of division. Otherwise, this section should be removed, as it is not conclusive.

Unfortunately, we have been unable to identify published markers that are asymmetrically segregated in mitotic cortical cells and can be visualized by immunofluorescence. However, to make this part of the manuscript more conclusive, we have analysed the division angles of mitotic cells using GFP, DAPI and the mitotic marker PH3 in the revised version. We have also added more cells to the analysis and replaced the previous low-resolution images of mitotic cells with higher resolution images (Fig. 7h). In addition, we focus our analysis on control, CyclinB1 and shRNA-CyclinB1/2 electroporated cells in the revised manuscript.

- 4) *The authors should elaborate a bit more as to the mechanism by which CyclinB1 regulates notch signaling. What is the mechanism by which notch signaling is regulated by Cyclin B1.*

In the manuscript, we show that CYCLIN-B1 can activate similar gene sets to those enriched in the multipotent progenitor populations we isolate and characterize. We also show that CYCLIN-D1 can suppress these in a reciprocal fashion. One such group of genes is involved in Notch signalling. Although we do not know the mechanism behind the regulation of these genes, it is interesting that CYCLIN-D1 has previously been shown to regulate Notch at the transcriptional level (Bienvenu et al. 2010). Thus, the finding that CYCLIN-B1 efficiently suppresses CYCLIN-D1 expression (Supplementary Fig. 11b) suggests that an indirect mechanism is likely to influence the expression of Notch signalling target genes. We have discussed this issue at p.22 of the revised manuscript.

Apart from modulating the kinase activities of CDKs, CYCLIN-D1 has also been found to play a role in the nucleus, whereby it regulates the transcription of genes such as NOTCH1, by attracting and regulating the activity of coactivators and corepressors^{22,53}. Whether CYCLIN-B1 acts via a similar mechanism to regulate genes implicated in stem cell maintenance is not currently established. However, we found that CDK1 is partly dispensable for CYCLIN-B1s ability to maintain cells in a RGC state. Thus, though CYCLIN-B1 is likely to act via CDK1 in regulating RGC differentiation, one interpretation of these results is that it may also act in a non-canonical, kinase independent manner (Fig. 9c). We further demonstrate that one important mechanism by which CYCLIN-B1 regulates the timing of corticogenesis is by upregulating the expression of genes involved in NOTCH signaling. While we can only speculate how CYCLIN-B1 mediates this function, it is notable that high levels of CYCLIN-B1 reduce the expression of CYCLIN-D1, which has been shown to modulate the expression of NOTCH by interacting with its cis-regulatory regions directly⁵⁴.

Referee #2

Specific comments

- 1) *For the tSNE figures in Figure 1 and Supplementary Figure 1, the way the authors perform dimension reduction, data visualization, and cell filtering is bizarre. Generally, for a high-quality single-cell RNA seq analysis, radial glia, IPCs, interneurons, excitatory neurons, and microglia should be readily identified and separated by an unbiased dimensional reduction using the most variable genes. The fact that the authors fail to separate these cell types using this way suggests either low-quality data or a too small sample size. The authors end up using a defined set of genes to perform dimension reduction which makes the figures biased towards the hypothesis.*

We can understand the reviewer's thoughts with regard to the inability of variable genes to extract different cell types from the data set and agree that this is an issue of their sample size. Our data set is dominated by neural cells at different phases of the cell-cycle and of different maturation stages. These specific cellular stages are defined by thousands of genes and represent the vast majority of the variable genes across our data set. Since variable genes scripts extract the primary drivers of variation across a data set, these are often incapable of segregating rare cell types when they are defined by relatively few genes.

In order to cluster and remove these cell types, we have therefore analyzed our data set based on the genes defining them. Previous work from the Linnarsson lab (Zeisel et al. 2015) has demonstrated that these specific cell types could be present in our data set and also allowed us to identify the genes defining them. Only by looking specifically at this limited set of 329 genes, as opposed to the thousands of variable genes, could we reliably cluster the astrocytes, interneurons and immune cells in our data set and identify them by their WGCNA gene set enrichments.

- 2) *The lineage analysis in Figure 2 is also strange. First of all, a tSNE plot is not most ideal to visualize lineages. Have the authors tried UMAP? UMAP is supposed to perform better in terms of consistent and meaningful organization of cells and to better visualize lineages. Also, instead of visually identify the differentiation trajectories on the tSNE plot, the authors should do this in a more sophisticated way using Monocle DDRTree, Slingshot, or RNA velocity.*

Although there are several packages available for dimensionality reduction and clustering, these work on very similar principles to the method we employ. Importantly, our visualizations and clustering are not based purely on tSNE plotting, but instead on a nearest neighbour network, weighted on the relative Euclidean distances between cells within the tSNE data. This means that any two cells that are connected within our graph share a closer relationship to each other than to all the other cells in our data set.

Since neurogenesis carries such a deep transcriptional footprint and cortical lineages are so closely related, with extensive co-expression of lineage determinants (Yuzwa et al. 2017; Cell Reports), it is striking that we find so many unconnected cells at the same stage of differentiation, which are instead more extensively connected

within the lineages we identify (Fig. 2a, b). This finding demonstrates the applicability of our bioinformatic method, which has also previously been shown to expose lineage relationships between cells within the developing ventral midbrain (Kee et al., 2017; Cell Stem Cell).

Regardless, we agree that it is important to show that the lineage relationships we identify can be found using other bioinformatic approaches. Thus, we have applied Monocle to determine a temporal order in our single-cell data. Using an instructive set of previously defined cortical layer identity genes (Zeisel et al. 2015) we readily define two distinct differentiation trajectories. A drawback with this analysis is that the defined trajectories only include cells from E15.5 and E18.5 cortices and thus could not define early differentiation decisions. However, Monocle's pseudotime of the cells did nicely confirm the maturation stage order of cells originally established in Supplementary Fig. 2 (R-squared value 0.97). In the revised manuscript these new data are presented in Supplementary Fig. 2 and discussed on p. 6.

- 3) *Also in Figure 2, when the authors define early and late differentiation trajectory, they based that on the expression of deep-layer and upper-layer genes. There are two problems with this. First, as mentioned in my second point, the authors should not assume a lineage relationship simply based on the proximity of clusters on a tSNE plot. For instance, it could be that cluster 2' gives rise to both clusters 3 and 3'. Second, the expression of the layer markers should be analyzed in each cluster separately, not in the whole "trajectory". Otherwise, the higher expression of these markers in more mature neurons will dominate the results and lead to wrong conclusions for the progenitor cells.*

As outlined above, the reason for our conclusions on cortical lineage relationships are based on the nearest neighbour connections between cells, and not their physical proximity, as in normal tSNE clustering. Furthermore, the definition of temporally distinct differentiation trajectories is also based on the age of the different clusters (Fig. 2b). Thus, it is unlikely that cluster 2' give rise to both cluster 3 and 3', as the cells of cluster 3 are of a younger embryonic age compared to cluster 2' (see Fig. 2b).

However, as the extensive co-expression of temporal lineage determinants within the cortex are gradually resolved when cells commit to a single fate, it may be true in the case of some genes that the difference in lineage determinant expression between trajectories is due to their expression in more mature neuronal cells. To better illustrate the connection between lineage specific gene expression in neurons and progenitors, we have produced a graph showing the sum correlation of each cell to the deep- and upper-layer trajectory gene profiles, which are presented in Supplementary Table 4. This data shows that the progenitors within each trajectory express the genes defining their own lineage neurons more highly than those defining the opposite trajectory, and thus provides important support for our lineage determinations. These new data are presented in Fig. 2e and discussed on p. 8.

Moreover, by correlating each cell in the tSNE-NN map to deep- and upper-layer neuroblast gene profiles (Supplementary Table 4), we found that progenitors within each trajectory expressed the genes defining their own lineage's neurons more highly than those defining

the opposite trajectory (Fig. 2e). Thus, the early and late differentiation trajectories identified appear to represent cortical cells undergoing deep- and upper-layer neurogenesis, respectively.

- 4) *The authors describe the use of a directed approach in order to examine the lineage relationship between cells. To do so, they use Prom1+ as a sole marker to FACS-sort committed and uncommitted progenitor populations from different cortical developmental timepoints for downstream bulk RNA-seq. Why was Prom1 chosen for this analysis? Why were different timepoints used to sample Prom1 negative and positive populations (E11.5 and 15.5 for Prom1+ and E12.5 and 16.5 for Prom1-)? Dissecting the intricate trajectory relationships of progenitor cells based on bulk-RNA sequencing of cells sorted by a single marker (at two timepoints) seems highly biased. The authors should explain the rationale for this approach.*

Prom1 was used as a progenitor marker for sorting because it is an established marker, with robust kits available for magnetic bead sorting. The different time points were used, as we wanted to capture both progenitors (PROM1⁺) and their daughter cells (PROM1⁻) from the deep- and upper-layer lineages. We felt that the neurons present at E12.5 and E16.5 would logically represent those born from E11.5 and E15.5 progenitors, respectively.

The aim of these experiments was to reveal lineage non-specific genes defining cortical neurogenesis. To achieve this, our strategy was to look for genes differentially expressed in neurons (PROM1⁻) and progenitor cells (PROM1⁺) (Deseq2 comparisons). Moreover, to remove lineage specific genes, we only considered genes differentially expressed by neurons and progenitors in both deep- and upper-layer lineages. These differentiation genes were then combined with the layer identity genes derived from (Zeisel et al. 2015), in order to inform our tSNE with both cortical lineage and neurogenesis components.

We understand that the way we had explained our thought process caused confusion and have changed the text describing this experimental strategy on p.6 of the revised manuscript.

- 5) *The authors hypothesize that the uncommitted E11.5 cells represent multipotent RGCs with the potential to generate both deep and upper-layer neuronal lineages. To test this, they chose cell-surface proteins enriched in upper-layer gene sets for FACS-sorting. It seems that these genes are also highly expressed in E11.5 newly committed upper-layer neurogenesis clusters (Fig. 2a, b; Fig. 3d). How can the authors be sure that FACS sorting is not selecting E.11.5 cells that are already committed rather than uncommitted E.11.5 cells as stated? These experiments show the propensity of E.11.5 cells to generate upper layer neuron identity at the expense of deep layer identity in vitro but have not shown the reciprocal experiment (generation of deep layer identity at the expense of deep-layer). Thus, there is a possibility that the in vitro conditions favour the generation of upper-layer identity regardless of cell potential.*

A progenitor's degree of commitment to different lineages is an important topic in our manuscript. As the referee points out, the cell-surface proteins used to isolate

multipotent progenitors are indeed expressed in cells that will contribute to upper-layer neurogenesis (e.g. Fig. 2a, b). However, the cells that the referee refer to are from E13.5 and E15.5 cortices. At E11.5 these markers are mainly enriched in uncommitted cell clusters (e.g. Fig. 2a, b). Thus, we do not agree that our FACS experiments isolating GPC6⁺, HMMR⁺ or EDNRB⁺ cells at E11.5 select for cells committed to upper-layer neurogenesis, but rather enrich for uncommitted progenitors (Fig. 4c, d; Supplementary Fig. 4m-p).

Moreover, as suggested by the referee we have performed the inverse experiment by targeting SLC1A5 and EFNA5, which are enriched on E11.5 cells committed to deep-layer neurogenesis (Fig. 2a, b; Fig. 4a). As demonstrated with our in vitro differentiation experiments these E11.5 cells are biased toward deep-layer neurogenesis at the expense of upper-layer neurogenesis (Fig. 4e, f; Supplementary Fig. 5f-h). Finally, it is important to point out that since our unsorted differentiation experiments performed at E9.5 (consisting of multipotent neuroepithelial cells) displayed a balanced generation of deep- and upper-layer neurons, we feel that our in vitro differentiation system as such does not favour the generation of deep- or upper-layer neurons (Fig. 3a-d).

- 6) *In Figure 4, are SOX2 levels of HMMR⁺, GPC6⁺, and EDNRB⁺ cells higher than SLC1A5⁺ and EFNA5⁺ cells? If SLC1A5⁺ and EFNA5⁺ cells are IPCs committed to deep-layer neurons, have the authors checked if they are TBR2⁺?*

Unfortunately, since we used a Sox2-eGFP heterozygous mouse for these experiments, in order to ensure we only sorted progenitors, Sox2 RPKMs are a compromised read out. However, to meet the referees request, and to show that HMMR⁺, GPC6⁺ and EDNRB⁺ cells express stem cell genes, while SLC1A5⁺ and EFNA5⁺ cells express higher levels of differentiation genes, we have added a panel to Supplementary Fig. 4 in the revised manuscript, which includes the RPKMs for the genes Notch1, Sox1, Neurod6 and Sox11 of each sort positive population. These new data are presented on p. 11;

To further characterize the identified populations of RGCs and lineage-committed progenitors, we next performed bulk RNA-sequencing on these cells directly after their isolation (Fig. 4b). Hierarchical clustering and differential gene expression analysis revealed substantial molecular similarities between HMMR⁺, GPC6⁺ and EDNRB⁺ cells, which expressed genes associated with RGCs (Supplementary Fig. 7a-d), and separated these from lineage-committed EFNA5⁺ and SLC1A5⁺ cells (Fig. 5a, b).

- 7) *Also in Figure 4, the authors should be clear about the terms they are using to describe cell populations. In the present manuscript, they are very vague and confusing. For example, on page 11, lines 2-3, the authors write "...both multipotent and committed progenitors (respectively referred to as RGCs and IPCs from this point on)...". Are committed progenitor cells all IPCs? How about older radial glia cells that only contribute to upper-layer neurogenesis? The authors should be very clear about the lineage (deep-layer) and the developmental time point (E11.5) of the committed*

progenitors they are referring to. They should also extend this clarification to the following paragraphs.

We appreciate this comment, as the subtleties of cellular commitment are an important topic in this paper. We obviously over-simplified these issues by using the established terms in the field of IPCs and RGCs. However, in the revised manuscript, we have now taken the reviewer's suggestion and used terms such as lineage-committed progenitors more widely, while using the term IPCs only when referring to TBR2⁺ cells. As suggested, we have also made sure to be specific about the lineage and developmental time point in all cases, in order to avoid any confusion.

- 8) *From Figure 4 back to Figure 1, if the E11.5 deep layer committed progenitor cells are indeed IPCs, why are they not EOMES/TBR2 positive in Figure 1? Does it infer the low quality of the single-cell RNA seq data?*

When RGCs commit to differentiation, they execute a cascade of events that is associated with the loss of NOTCH signalling and activation of NEUROG2. This is followed by the upregulation of more definitive markers, such as TBR2, TBR1 and cortical layer markers. Thus, cells commit to differentiation long before upregulating TBR2, but these cells lack a widely accepted definitive name in the literature until they become TBR2⁺ IPCs.

As stated above, the previous manuscript attempted to skate over this issue by using the term IPC. On the reviewer's advice, we have used the more accurate term of lineage commitment in the revised manuscript, while only referring to TBR2⁺ cells as IPCs.

Minor Points

- 9) *In figure 6c, 6e and Supp. Fig7 c,e,f. for some images it seems that SATB2 or POU3F2 are expressed from the VZ all the way through to the upper cortical layer. At E18.5 they should be restricted to the CP region. Is this an antibody specificity issue or are regions of cortex mislabeled?*

Although not widely discussed in the literature, at E18.5 SATB2⁺ and POU3F2⁺ are actually found distributed throughout the VZ, SVZ and CP. We have attached figures from the Allen Brain Atlas, which demonstrate that the pattern we see is observed by others (See "Figure for inspection by referee #2").

- 10) *There are several citation oversights. The function of cyclin D in promoting neurogenesis has been reported before by the Anderson lab 1. The ability of NOTCH intracellular domain to keep progenitors as proliferating radial glia cells has also been reported previously 2. These findings should be properly cited.*

We apologize for these oversights. These are important papers that are referenced and discussed this in the revised manuscript.

The data originating from the Anderson lab (Lukaszewicz, 2011) is in revised manuscript discussed on p. 4

... CYCLIN-D1 has been implicated in controlling the onset of neurogenesis by promoting the formation of IPCs^{17,18}, whereas the ability of CYCLIN-D1 to promote neurogenesis in the spinal cord can be dissociated from its cell-cycle function¹⁹.

and p 20;

For instance, CYCLIN-D1 has been previously described to induce an IPC state in the cortex^{17,18} and promote neurogenesis in the spinal cord¹⁹, which is consistent with our bioinformatic and functional experiments.

The data originating from the Saito lab (Mizutani, 2005) is discussed at p. 20;

Notably, although forced expression of CYCLIN-B1 prevented cortical RGC differentiation, it did not preserve their multipotent cell-fate competence. Instead, the electroporated cells underwent normal lineage progression and committed to later formed cortical lineages in an appropriately timed manner. Moreover, we and others have obtained similar results by inhibiting cortical neurogenesis through NICD overexpression⁵⁰.

Referee #3

Concerns

- 1) Page 6 Line 6: "To achieve this, we performed bulk RNA sequencing on PROM1+ progenitors from E11.5 and E15.5 cortices and differentiating PROM1- cells from E12.5 and E16.5 cortices." Why did the authors do these experiments at different ages? They could have taken PROM1+ and PROM1- cells from the same cortices at E11.5 and E15.5 for instance.

Prom1 was used as a progenitor marker for sorting because it is an established marker, with robust kits available for magnetic bead sorting. The different time points were used, as we wanted to capture both progenitors (PROM1⁺) and their daughter cells (PROM1⁻) from the deep- and upper-layer lineages. We felt that the neurons present at E12.5 and E16.5 would logically represent those born from E11.5 and E15.5 progenitors, respectively.

The aim of these experiments was to reveal lineage non-specific genes defining cortical neurogenesis. To achieve this our strategy was to look for genes differentially expressed in neurons (PROM1⁻) and in progenitor cells (PROM1⁺) (Deseq2 comparisons). Moreover, to remove lineage specific genes, we only considered genes differentially expressed by neurons and progenitors in both deep- and upper-layer lineages. These differentiation genes were then combined with the layer identity genes derived from (Zeisel et al. 2015), in order to inform our tSNE with both cortical lineage and neurogenesis components.

We understand that the way we had explained our thought process caused confusion and have changed the text describing this experimental strategy on p.6 of the revised manuscript.

- 2) *Page 6 Line 12: "As visualized in the tSNE-derived nearest neighbour map, this analysis effectively separated cells according to their age and expression of the progenitor marker *Prom1*, the intermediate progenitor marker *Eomes* and the neuronal marker *Dcx* (Fig. 1c, d)." Why was bulk RNA-seq used to identify these markers? There are already known markers.*

As described above (comment 1) bulk RNA-seq was used to define genes representing differentiating cortical neurons, which together with cortical lineage genes were used to inform our tSNE. In Fig. 1d we visualize *Prom1*, *Eomes* and *Dcx* to demonstrate that the cells in our tSNE are separated according to known differentiation markers and maturation values (Fig. 1e).

- 3) *The method used to define the maturation stage value is unclear. The authors should provide a clearer and more comprehensive explanation. "Maturation stage assignment". The authors should provide references or list the genes used. "PC1 and PC2 were then rotated such that the number of detected genes correlated directly with PC1": What does it mean? What is PC1 (=time?)? This should be carefully justified.*

In order to gain an independent and linear measure of each cell's differentiation state, we started by performing PCA on our cells using a set of previously published differentiation genes (Kee et al, 2017). This allowed us to avoid using the genes we used to inform the tSNE, to also assess it. Although we had mistakenly omitted this reference, it has now been added to our revised Methods section.

Our experience working with developmental single cell RNA-seq data sets has demonstrated that the number of genes detected and differentiation are generally the first and second most significant sources of variation within a data set. Thus, these two factors dominate principal components 1 and 2. However, since PCA is an unbiased analysis, they are blended between principal components 1 and 2. In order to isolate a linear scale of cell maturation we rotated PC1 and PC2, using the same prcomp R package used for the PCA, until PC1 was perfectly aligned with the variation in genes detected. We illustrate this rotation and its alignment with detected genes in Supplementary figure 2d.

In order to further clarify the meaning of the maturation stage value, we have now also run our data set through the Monocle package. Monocle Pseudotime is a well-established maturation stage measure and correlated strongly ($r^2 = 0.97$) with our own maturation stage measure. Moreover, we now also use Monocle to confirm the branching of our lineages. These new data are presented in (Supplementary figure 2a and 2i).

- 4) *"Furthermore, E9.5 cells displayed significantly greater gene expression correlations to the E11.5 cells from the Infomap clusters lacking maturation stage pairs than to E11.5 progenitors newly committed deep- or upper-layer neurogenesis (Fig. 2a, b; Fig. 3d)." What is the relevance of this statement?*

The aim of this section is partly to provide evidence that cell-clusters lacking maturation stage pairs (see Fig. 2b) represent uncommitted cortical progenitors. We show this by demonstrating that E9.5 cells are multipotent in a differentiation assay in vitro. We also show in Fig. 2d, that the E11.5 and E13.5 cell clusters lacking maturation stage pairs are more similar to the multipotent E9.5 cells than cells newly committed to deep- and upper-layer neurogenesis. Thus, the relevance of the sentence the referee is referring to, is to state that cells lacking maturation stage pairs are molecularly more similar to E9.5 cells than cells newly committed to deep- and upper-layer neurogenesis.

Moreover, the sentence referred to has in the revised manuscript been changed to;

Furthermore, E9.5 cell transcriptomes displayed a significantly greater correlation to the E11.5 cells lacking maturation stage pairs than to E11.5 or E13.5 progenitors newly committed to deep- or upper-layer neurogenesis, respectively (Fig. 2b; Fig. 3d).

- 5) *Regarding Fig. 3, description of the bioinformatic analysis is very convoluted. Why was WGCNA used, what are the gene set 1 and 2? The authors should refer to a Sup Table here. In f, where do Cdca2 and Orc1 emerge from?*

WGCNA is a package designed to identify genes with similar expression patterns across entire data sets. Deep- and upper-layer trajectory gene sets are clusters of genes that show enriched expression along the early and late trajectories we define in Fig. 2. This was an invaluable tool for us since, in an unbiased fashion, it allowed us to identify the proteins we used to sort deep- and upper-layer committed progenitors, as well as to uncover the enrichment of cell-cycle phase-specific genes in the different trajectories. *Cdca2* and *Orc1* are classical genes involved in mitosis and DNA-replication and were found in upper layer gene set 1 and deep layer gene set 1, respectively. In order to clarify this, we have now altered the text on p. 9 describing Fig. 3 to read;

To uncover groups of gene that describe cortical cells as they commit to deep- and upper-layer trajectories, we next performed WGCNA on genes differentially expressed between Infomap clusters of similar maturation stages. This approach revealed groups of genes expressed during the earliest stages of commitment to cortical neurogenesis. To visualize the different gene sets, their average expression scores were plotted on the tSNE-NN map and the in vivo mRNA expression patterns of representative genes were examined (Supplementary Fig. 3a, b)³¹. Interestingly, the gene sets enriched in cells at early commitment stages to deep- or upper-layer neurogenesis (deep-layer trajectory gene sets 1-2 and upper-layer trajectory gene sets 1-2; Supplementary Fig. 3a, b) converged in uncommitted E11.5 cells, which co-expressed genes

enriched in both differentiation trajectories (Fig. 3e, f; Supplementary Fig. 3a, b).

- 6) *Slc1a5⁺, Efna5⁺, Sox2^{low} cells are IPs. Given the tSNE in Fig. Sup. 4e, f, how many of the collected cells co-express Slc1a5 and Efna5 in the scRNAseq? Given the tSNE in Fig. 1 representing Eomes expression (a classical IP marker) it seems that most of these cells do not express Slc1a5 and Efna5; how do the authors explain this? What is the number of cells that co-express these markers?*

Considering co-expression of *Efna5* and *Slc1a5* (over RPKM 1) we found that 58% of cells expressing *Efna5* also express *Slc1a5*. In fact, the overlap enrichment between *Slc1a5* and *Efna5* is similar to that of our uncommitted cell markers, and much higher than for markers that should not be widely co-expressed, such as *Prom1* and *Dcx* (See Fig inspection Ref #3).

As the referee points out, the *Slc1a5⁺* and *Efna5⁺* cells in our data set do not generally express the intermediate progenitor cell marker *Eomes*. Thus, our data support the idea that cortical progenitor cells commit to neurogenesis before they upregulate *Eomes* expression, for instance by reducing Notch signalling and upregulating *Neurog2*.

- 7) *Regarding Fig. 5 and the gene set enrichment analysis, it is hard to understand how the authors found the GO. Are they at the top of the list? It might be nice to have a ranked list of GO. In 5c, the displays are just impossible to understand for non-specialists. The authors should re-think this and other panels.*

Since our GO analysis is based on comparisons between several different uncommitted and committed cell populations, the terms that we chose to focus on were based on the GO terms that consistently reoccurred in our comparisons. Although we used several GO databases to get a full picture of the pathways enriched in our different sorted populations, we have provided a full list of all of the GSEA Hallmark GO terms that are enriched in both *SLC1A5⁺* and *EFNA5⁺* cells when they are compared to both *HMMR⁺* and *EDNRB⁺* cells in (Fig inspection Ref #3). Although some other stem cell pathway or stress response terms could have been chosen for display, we feel that the pathways we show are a good representation of our findings.

As for the GSEA displays, these are now commonly shown in many publications due to the amount of information they contain. GSEA is a unique tool, since it does not impose cutoffs on the significantly differentially expressed genes it includes in its analysis and this allows the calculation of an enrichment score based on how differentially expressed all the genes in a GO term arm. The graphs themselves show the ranked fold change of every gene in a specific GO term for any given comparison, which is important information for judging the significance of a result.

- 8) *Page 11 Line 20: "To proceed, RGCs and IPCs sorted from E11.5 cortices were labelled with propidium iodide (PI) and their cell cycle phase enrichments were determined*

using FACS.” The authors should explain in the text how PI works (that the intensity of the fluorescence detected by the FACS will determine the cell cycle phase). Related to Fig 8b. Why show Neurog2 and Tubb3? There is no reference of this result in the text (see Page 15).

In the revised manuscript, p.12, we have better explained the rationale of using PI and FACS to determine cell-cycle phase enrichments;

To proceed, the DNA of RGCs and lineage-committed progenitors, sorted from E11.5 cortices, was labelled with propidium iodide (PI), so that their DNA content, and thus cell-cycle phase enrichment, could be determined using FACS.

The reason why we display the expression of Dll3, Neurog2 and Tubb3 in Fig. 8b is that the expression of these genes are negatively regulated by Notch (in a cell autonomous manner). In line with this, when misexpressing CYCLIN-B1 in cortical cells we found a negative correlation between these genes and Notch1/2. In the revised manuscript, we have clarified this issue and properly referenced the data on p.16.

9) *Related to Fig.9c. IPCs communicate with RGCs through NOTCH signaling. Which data show this?*

That committed cells in the cortex communicate with RGCs through NOTCH signalling has been suggested previously (e.g. Kawaguchi et al, 2008; Nelson et al, 2013; reviewed by R. Hevner, 2019). Moreover, this is in line with our findings that high levels of CYCLIN-B1 upregulate Notch1/2 whereas misexpression of CYCLIN-D1 leads to a suppression of these genes.

In the revised manuscript we are avoiding the term IPC unless the cells are TBR2+. Accordingly, in the revised summary Fig. 9c we use the term “lineage-committed progenitor” instead of IPCs.

Other concerns

1) *Page 3 Line 7: “and finally cortical glia, 1.”: please remove “,”.*

Corrected

2) *In the introduction, a comma is missing between references in the text.*

Corrected

3) *Page 14 Line 12: “(Fig. 7d,)”. e is missing.*

Corrected

4) *In Fig. 8c and Sup Fig. 10, it would be nice to show the images for GFP control.*

In the revised manuscript, we have now included GFP control electroporations in Fig. 8c, Fig. 9a and Supplementary Fig. 12 (Former Supplementary Fig. 10).

5) *Discussion Page 19: "Heterochronic transplantation...", "Thus, these results are consistent with a model by which a population of multipotent RGCs becomes progressively fate restricted with time." The authors should present and discuss the recent findings of Oberst et al, 2019 in this context. Also Telley et al, 2019, Pilaz et al, 2016, Pilaz et al 2009 are important contributions which are missing.*

We apologize for these oversights. In the revised manuscript these papers are referenced and discussed. Telly et al, 2019 and Oberst et al, 2019 are referenced at p. 20-21;

Although it is not fully understood how fate progression of RGCs is regulated, cell-cycle progression has been demonstrated to be dispensable for these cell-fate transitions⁵¹. However, the loss of Eed, the regulatory subunit of PRC2 that regulates chromatin accessibility, has previously been shown to accelerate fate competence progression of RGCs ⁵². Moreover, heterochronic transplantation of RGCs into younger cortices causes these progenitors to re-enter a competence state typical of RGCs contributing to deep-layer neurogenesis ⁵³. One possible explanation for this temporal plasticity is the high levels of Wnt signalling components in younger cortices ⁵³.

Pilaz et al, 2016 is references at p. 20;

However, in contrast to the above findings it should be noted that pharmacological induced prolongation of the M-phase has been shown to promote RGCs to commit to neurogenesis⁴².

Pilaz et al, 2009 is referenced at p.4;

For example, by regulating the length of the G1 phase in the cortex, CYCLIN-D1 has been implicated in controlling the onset of neurogenesis by promoting the formation of IPCs^{17,18}, whereas the ability of CYCLIN-D1 to promote neurogenesis in the spinal cord can be dissociated from its cell-cycle function¹⁹.

Pilaz et al, 2009 is referenced at p.21;

For instance, CYCLIN-D1 has been previously described to induce an IPC state in the cortex^{17,18} and promote neurogenesis in the spinal cord¹⁹, which is consistent with our bioinformatic and functional experiments.

REVIEWERS' COMMENTS:

Reviewer #1 (Remarks to the Author):

The authors have addressed my concerns. I am happy with the revised manuscript.

Reviewer #2 (Remarks to the Author):

I find it strange that the authors could not cluster rare populations (interneurons, astrocytes and microglia) using top variable genes. It seems that this is likely due to a technical bias and small sample size (less abundant cells with more complex and arborized morphology are less likely to be sampled using this technique).

I think that the rationale behind the use of prom1 +/- FACS sorted cells for downstream bulk-RNA seq is better explained. Considering all the data presented and the thorough functional validations performed, I think there is enough evidence support for two progenitor populations giving rise to upper and deep-layer neurons.

It is now clearer why the authors chose to use tSNE-NN as it separates the RGC population better for them to see the two trajectories. I am more convinced that the two trajectories they presented are real.

Reviewer #3 (Remarks to the Author):

We thank the authors for addressing our questions.

2 points:

1- In comment 6, the authors state "Thus, our data support the idea that cortical progenitor cells commit to neurogenesis before they upregulate Eomes expression." This is not mentioned in the manuscript and would benefit from being explicitly stated.

2- Comment 7, GSEA Hallmark GO terms do not correspond to the ones cited in Figure 5. Please correct or explain.

Page 6 Line 22: "tSNE-NN map map": correct.

REVIEWERS' COMMENTS:

Reviewer #1 (Remarks to the Author):

The authors have addressed my concerns. I am happy with the revised manuscript.

Reviewer #2 (Remarks to the Author):

I find it strange that the authors could not cluster rare populations (interneurons, astrocytes and microglia) using top variable genes. It seems that this is likely due to a technical bias and small sample size (less abundant cells with more complex and arborized morphology are less likely to be sampled using this technique).

I think that the rationale behind the use of prom1 +/- FACS sorted cells for downstream bulk-RNA seq is better explained. Considering all the data presented and the thorough functional validations performed, I think there is enough evidence support for two progenitor populations giving rise to upper and deep-layer neurons.

It is now clearer why the authors chose to use tSNE-NN as it separates the RGC population better for them to see the two trajectories. I am more convinced that the two trajectories they presented are real.

We are happy that Reviewer #2 now find our findings convincing.

Reviewer #3 (Remarks to the Author):

*We thank the authors for addressing our questions.
2 points:*

1- In comment 6, the authors state "Thus, our data support the idea that cortical progenitor cells commit to neurogenesis before they upregulate Eomes expression." This is not mentioned in the manuscript and would benefit from being explicitly stated.

We agree that this is an important comment. In the revised Discussion at page 19 we now state;

While, the expression of Efn5 and Slc1a5 demarks committed progenitors, it is interesting to note that cortical cells expressing these genes are generally devoid of Eomes expression (compare for instance Fig. 1 and Supplementary Fig. 4). Thus, our findings support the notion that cortical progenitor commit to neurogenesis before they upregulate the expression of the IPC marker Eomes, for instance by reducing NOTCH signalling and upregulating Neurog2.

2- Comment 7, GSEA Hallmark GO terms do not correspond to the ones cited in Figure 5. Please correct or explain.

The ontology terms presented to reviewer 3 were taken from the hallmark gene sets, as these contains the smallest set of terms and was meant to illustrate why we chose to present the terms that we ultimately did. Gene ontology terms involved in mitosis, TGF β and Notch signalling arose repeatedly in all the gene sets we analyzed for genes enriched in GPC6, EDNRB and HMMR positive cells. Likewise, G1-S-phase, differentiation and MYC targets arose repeatedly in all the gene sets we analyzed for genes enriched in SLC1A5 and EFNA5 positive cells. Thus, we selected specific terms from the different gene sets we looked at to illustrate these term enrichments.

Page 6 Line 22: "tSNE-NN map map": correct.

In the revised manuscript this error has been corrected.